



# Using geosciences and mythology to locate Prospero's Island

Tiziana Lanza

Istituto Nazionale di Geofisica e Vulcanologia (INGV) Rome, Italy

*Correspondence to*: Tiziana Lanza (tiziana.lanza@ingv.it)

**Abstract.** *The Tempest*, the last work entirely attributed to William Shakespeare, has been subject to many studies and interpretations, ranging from adventure, Shakespeare's biography to colonialism and cultural revolution and is studied in this paper in the context of natural hazard. The play tells about a magician, Prospero, and his daughter who are shipwrecked on an unknown island where they encounter

strange creatures and beings. It is the same island where King Alonso and his crew shipwreck twelve years later. But is it a fantastic island or was the author inspired by a real place? Literary scholars have done several hypotheses through the years based on historical sources. Indeed, analysing the verses describing the tempest in the light of geosciences and mythology supports the hypothesis that Shakespeare was inspired by the Mediterranean. We don't intend to identify the island, we believe the

island is a polyhedral place both from a philosophical-political and from a geographical-environmental point of view. Nevertheless, some verses of the play suggest volcanism placing the island in the Sicilian sea. This underlines once again how deep the knowledge of the playwright about Italy was. It also suggests that this part of the Mediterranean was known, at the time of Shakespeare, as the theatre of phenomena originated by the volcanism of the area. One implication would be that the Shakespeare

could have used sources precious to reconstruct geological events occurred out off the Sicilian coast.

## 1 Introduction

In recent years a new trend towards the re-unification of the two main streams of culture, the humanistic and the scientific, is becoming more evident year by year. Scientists and artists co-create projects to address issues of societal importance in a holistic way and to improve on science communication. Earth

scientists, in particular, are familiar with studying historical records and literary accounts, mythology, story telling in addition to geo-archaeological evidences to reconstruct a time line of historic



catastrophic events such as volcano outbreaks, floods, storms, etc. Very often seismologists depend on historical sources to understand the present geology of a site, which may have been shaped by several and reiterated seismic events. Even with the evolution of technology that has brought geo-scientists new

sophisticated methods of investigation, modern seismology cannot do without a deep immersion into historical and literary accounts for calculating, for instance, the return period of an earthquake. The Istituto Nazionale di Geofisica e Vulcanologia (INGV) contributed significantly to this by collecting in an original volume the ancient earthquakes in the Mediterranean area up to the 10th century (Guidoboni et al. 1995). It also compiled a catalogue of strong Italian earthquakes (Boschi, E. et al 1995). The

catalogue is kept up to date and recently has been extended to the large earthquakes (6.0-6.9 Magnitude) of the Mediterranean area. Details about the new updated front-end have recently been published by Guidoboni, E. et al. (2019). The Catalogue is a direct example of how literary accounts ranging from historical sources until the most recent chronicles can help to document also minor, little details, that can make possible to better understand an earthquake of the past in terms of Magnitude, intensity and

social impact. Even for present earthquakes our Institute, since many years, has organized data collection from citizens providing details that complement the data collected through the scientific instrumentation (De Rubeis et al. 2015) in the same way as historic accounts fill the gap for past records.

Combining history, literature, geo-mythology and archaeology accounts play a fundamental role in

reconstructing past volcanic events and their impacts on society and environment. In the renown case of the 79 AD Vesuvius Eruption that destroyed Pompeii and Ercolano, archaeologists had the opportunity to reveal much about the life of the inhabitants thanks to the massive pyroclastic surges and ash fall deposits that buried the Roman settlements. Similarly, the accounts of Pliny the Younger, a Roman administrator and a poet, who witnessed and documented the catastrophic event (Jones 2001), were

particular important for geo-scientists. Pliny the Younger in his letter to Tacito referred of several earthquakes occurring the days before the eruption. The effects of these earthquakes are still visible in several buildings in Pompei and Villa Regina. The words of Pliny who was observing the event from Misenum at a distance of 21 km were also precious to reconstruct the eruptive cloud described as a "Mediterranean pine". While in the morning of the second day of the eruption Pliny the Younger



observed the development of pyroclastic flows descending down the flanks of Vesuvius and flowing on the sea. The description of Pliny the Younger fits well the geologic record of the eruption (Giacomelli et.al 2003).

Some events remain difficult to reconstruct because of a lack of observers. For example, G. Mercalli (1883) asserts that there are only few records about volcanic eruptions at sea, since the phenomena at the time could be witnessed predominantly by sailors. Mercalli reports some episodes that happened in the Sicily Channel including the event when the most popular Ferdinandea island emerged from the sea in 1831 (Fig.1). As we will show later, we are stunned by the way Shakespeare describes a natural phenomenon in a way so similar to the one described by Mercalli in his book published only a couple of centuries later.

## 2 Objectives and Methodology

This study aims to show how literature and geo-sciences can benefit of a mutual exchange in order to add new and fresh elements each on their own investigative area. In particular, in this paper we will consider the opposite case of what was previously described: we will use geosciences and geo-mythology to better understand a work of Art wordily renown as a masterpiece of William Shakespeare (from now on WS): *The Tempest* (Kermode ed. 1986)

In doing so, we intend to do a work of eco-criticism. It is our intention to collect all the indications that can help us to analyse all that in the play is connected to a real location in terms of an environmental and geophysical asset, using sources from geoscience studies, historical and others.

As a first step, we introduce the reader to the period of the English Renaissance, a time that is referred to as a time of scientific revolution and great scientific discovery and which greatly influenced WS.

In a second step we address the locations of WS's plays to introduce *The Tempest* and one of the unsolved questions among literary scholars: the location of Prospero's island.



Finally, taking into account both the geology off the Sicily coast and the mythology of the Mediterranean area, we propose an interpretation of Shakespeare's verses of *The Tempest* in a new and

so far never considered perspective.

**3 On WS in relation to this work**

**3.1 Main streams of culture in the English Renaissance**

In his book *Vorlesungen zur Philosophie der Renaissance* Ernst Bloch defines "hot" and "cold" the two

main Renaissance currents of thought. The "hot" includes pantheism, mysticism, astrology, alchemy and magic, while the "cold" Machiavellianism, empiricism, astronomy and physical-mathematical sciences (Bloch 1972). During the Renaissance, a new interest for the orphico-pythagorical tradition gave an impulse to the studying of the occult disciplines. The protagonist is the new Renaissance magus, astrologer and alchemist, owner of a deep knowledge and able to discover the secret processes

of nature with the intent to control it.

The Elizabethan Age gives an example of how, the emerging scientific disciplines like astronomy, chemistry, physics coexisted with the fashion for occultism, magic, cabalism, astrology and alchemy. A literary critic, C. Clark (1938) explains that even if there were sceptics and doubters because this was an

Age which saw the beginning of what are today established sciences – for instance the Copernican school of astronomy was questioning the rules of the astrologers while the pioneers of chemistry were challenging the claims of the alchemists – some of the neo-scientists adopted an attitude of extreme tolerance towards ideas they knew to be non-sense. The reason, concludes Clark, is that "dry scientific facts did not of themselves win that support and help without which progress was impossible".


During the Reign of James I, when *The Tempest* was for the first time performed, the interest for occultism had not yet faded away. The king himself was a lover of demonology. Nevertheless, we assist to a slow decline of the Renaissance magus. John Dee (1527 – 1608) is a clear example of it: he was a mathematician, astronomer, astrologer and occult philosopher, and an advisor of Queen Elizabeth I.

Accused several times of sorcery, he died in poverty under the reign of James I. What happened in





Dee's lifetime – write F. Yates (1975) – to his "Renaissance Neoplatonism" was happening all-over Europe, as the Renaissance turned into the darkness of the which-hunts. However, the occult disciplines gave their contribution to the development of thoughts reaching its climax in the intellectual revolution initiated by Francis Bacon, the English philosopher credited as developer of the new scientific method.

His works remained influential through the scientific revolution. Nevertheless, Bacon himself reinterpreted one of the main features of the magical-alchemical tradition: the philosophy of domain, a knowledge aiming at transforming nature (Rossi, 1968)

We don't intend to discuss in this paper the different hypothesis on the real identity of WS.

Nevertheless, to this respect we just mention the so-called Baconian theory of Shakespeare authorship. The so-called Baconians, among the other evidences, have also argued that Shakespeare's works show a detailed scientific knowledge that, they claim, only Bacon could have possessed.

This was the cultural period during which WS conceived the main character of *The Tempest*, Prospero.

It is difficult to give a precise definition of who Prospero really represents. He oscillates in between the old renaissance magus and the new philosopher/scientist that was showing up on the horizon. He is a magician, and some have also speculated that WS was inspired by John Dee in portraying him. However, the way Prospero speaks and behaves seems to recall the new empiric method Bacon was developing to observe nature. Not surprisingly, therefore, that some have speculated (Falk 2014) that

Prospero was inspired by the astronomer Tycho Brahe, and his island/observatory, Uraniborg. As pointed out by science journalist Dan Falk, Shakespeare lived during a remarkably eventful period in terms of celestial drama. Falk enumerates all these amazing events including passages of comets, solar eclipses, and moreover the appearance, of a bright new star in the constellation of Cassiopea, in November 1572. It was so bright that for several months it outshone even Venus. It was observed by

Digges in England, and monitored even more closely in Denmark by astronomer Tycho Brahe. Today the star is named "Tycho's star".





Behind Prospero, there is also his creator, WS himself. Not surprising therefore to assume that what in appearance should be considered the product of Prospero's potent art, able with his magic to provoke catastrophic events, becomes a careful account of natural phenomena.

Not only Prospero uses the adjective "rough" referring to his magic, commentators also remark the halo of ambiguity that concerns Prospero's books. We learn from his own words, that he is a sort of researcher rapt in secret studies (*Pros.* And Prospero the prime duke, being so reputed |In dignity, and for the liberal Arts| Without a parallel; those being all my study, | The government I cast upon my brother, | And to my state grew stranger, being transported | And rapt in secret studies I.ii. 72-77). In the
same tale addressed to his daughter Miranda during which she learns how they survived and reached the island, Prospero, mention Gonzalo's good hearth with respect to his loved books (*Pros.* … so, of his gentleness, | Knowing I lov'd my books, he furnished me | From my own library with volumes that | I prize above my dukedom I.ii 164-168). But, remarks Marnieri, when he solemnly pronounces his renunciation to magic art, he speaks about one "book" he will "drown" (*Pros.* But this rough magic | I
here abiure; and, when I have requir'd| Some heavenly music, - which even now I do, - | To work mine end upon their senses, that | This airy charm is for, I'' break my staff, | Bury it certains fathoms in the earth, | And deeper than did ever plummet sound | I'll drown my book V.i. 50-57). We may infer that Prospero is conscious of the new rational science which is becoming the dominating culture of the age (Marnieri 2013)

**3.2 Shakespeare's locations**

In fig. 2 the map shows the locations of the Shakespearean works. At a first glance, you can see that most of the plays are obviously located in the U.K. But it is immediately evident that also Italy is amongst the favourite countries by WS to set his plots. In fact, one third of the Shakespearean plays is located in Italy. There is a lot of literature about the interest that WS nourished for the country. He knew
so much about it, even in detail, that some recent studies speculated about the fact that the works of WS can be studied only to the light of his relation with John Florio the son of Michelangelo Florio born in Tuscany (Gerevini 2008). His family migrated to England at the time Italy was under the Spanish





domination to escape persecution as Jews. John Florio in the English cultural landscape, at the turn of the sixteenth and seventeenth centuries, was considered a different, an Italian emigrant of Jewish

origins, so inconvenient and looked upon with suspicion by many! He then decided to hide his own identity, as a play-writer, behind a co-operation with Will of Strafford, the actor William Shagsper of Stratford.

As we have already said, this is not the place to discuss the dispute on the real identity of WS. To the aim of this paper it is only important to remark that he was familiar with Italy, with its culture, with the

places where he located his work, and as we will show, also with the geophysical phenomena happening out of the Sicilian coast. Besides the roman plays, *Anthony and Cleopatra*, *Coriolanus*, *Julius Caesar* and *Titus Andronicus*, he located in Italy *Romeo and Juliet* (Verona); *The two Gentlemen of Verona* (Milan); *The Timing of the Shrew* (Pisa and Padua); *The Merchant of Venice* (Venice); *Othello* (Venice); *A Midsummer Night's Dream* (Mantova, Sabbioneta); *All's Well that Ends Well* (France and

Florence); *Much Ado about Nothing* (Messina); *The Winter's Tale* (Sicily); *The Tempest* (?). The last two are locate in Sicily (apart *The Tempest*).

We hypothise that also *The Tempest* was located in Italy, even if the official location of the play is simply "an island" for the following reasons: first, almost all the characters are Italians. The protagonist, Prospero, is the duke of Milan. The crew is composed by Alonso, King of Naples and Antonio his

usurper brother, and the rest of the crew are also Italian. The only stranger is Caliban, the slave of Prospero, and Sycorax his mother who comes from Algeri. Second, the story starts in Italy (as by the tale of Prospero) in Milan and ends in Milan as Prospero's words pre-announce. Thirdly, as we will show in this paper, there is evidence in the play itself that the island may be located somewhere in the Sicilian sea.






### 3.3 Introducing The Tempest

*The Tempest* is considered the last play of WS mainly because it is a container of all the themes the
playwright treated in his former works. It was written probably between 1610-1611 and performed for
the first time on 1 November 1611 at Court. It was published later in the *First Folio* of 1623 from an
edited transcript by Ralph Crane, the scrivener of the King's Man, the theatrical company to which WS
belonged for most of his career.

Here is a brief synopsis: Alonso King of Naples and his crew (from now on "The Court Party"),
composed by his son Ferdinand and his brother Sebastian, a wise counsellor, Gonzalo, and Antonio, the
Duke of Milan, are coming back to Naples from Tunis where they assisted to the King's daughter
wedding. They shipwreck and reach a mysterious island, where Prospero, the former Duke of Milan,
live with his daughter Miranda. In assisting to the  cast away from the shore, Prospero tells Miranda for
the first time  how they came to be on the island. Twelve years before, his brother Antonio has usurped
him the title of Duke of Milan. Fortunately, with the help of Gonzalo, he succeeded in escaping with a
small boat and his baby daughter, bringing also with him his books about magic. On the island, they
found Caliban, a deformed and savage creature that Prospero turned into his slave, and a spirit, Ariel
(Fig.3), who had been imprisoned in a tree trunk by a witch, Sycorax, Caliban's mother, who had then
died. Prospero rescued Ariel and with the help of magic obtained his devotion. All the ship's passengers
are unharmed and even their cloths are not wet or damaged. They  reached different parts of the island.
Alonso is searching for his son Ferdinando, and the reverse. While searching for his father, Ferdinando
meets Miranda and they fall in love at first sight. In the meanwhile, several conspiracies are taking place
on the island. Antonio and Sebastian plan to kill Alonso so that Sebastian can become king. Trinculo
and Stephano, the court jesters, are recruited by Caliban for helping him to overthrow Prospero.  Their
conspiracy is foiled by Ariel. Once the Court party is reunited by Ariel in front of Prospero's cell, he
renounces his magic and reveals himself. He forgives his brother and prepares himself to return to
Milan and to resume his dukedom. Ferdinando and Miranda are engaged. Everybody is forgiven and
sailors arrive announcing that the ship has not been wrecked after all, and is safely anchored off the



island. Ariel is set free and all ends in a final celebration crossed by an unexpected Prospero's melancholy.

### 3.4 Somewhere beyond the sea: Shakespeare's sources for the Tempest

While for most of his other plays, Shakespeare makes reference to sources that are more or less easy to to identify for *The Tempest* there is a never ended debate on the sources that inspired WS, especially the ones concerning the island. He used sources to build the philosophical and political asset of the play as for instance the Essay of Montaigne *Of Cannibals*, or, probably, Nicolò Macchiavelli, *The Prince*. The inspiration for the character of Prospero, the impeached Duke, may have been adapted from William

Thomas's *History of Italy* (1549) in which Prosper, the Duke of Genoa, was deposed in 1561 (Gerevini 2008). Possible Italian sources come from the Commedia dell'Arte (*Li Tre Satiri*, *Il Mago* and *La Nave*). Other sources are more concerned with geographical accounts, as the so-called Bermuda Pamphlets, and in particular, *A true repertory of the wreck and Redemption of Sir Thomas Gates Knight*, William Strachey, 1610 and *Discovery of the Bermudas otherwise called the "isle of Devils"*,

Silvester Jourdain 1610.

There is a lot in the play that can be connected to the mythology of the Mediterranean area. Classical sources are Ovid's *Metamorphosis*. Of not less importance is Virgil *Aeneid*, present in the play with precise references (see par. 4.3). Very probably, WS read also *The Geography* of Strabo. He may also

have used other sources on volcanic phenomena occurring at sea. Probably WS learned about it from the captains and the sailors he came in contact or maybe he read the board diary of the English vessels. But here we can just speculate, since an historical research on volcanic phenomena at sea occurred in the past have not yet been performed. Mercalli emphasised the importance of conducting such studies on the submarine eruptions occurred in the Mediterranean sea. He remarked:


"Yet for geology the study of underwater volcanoes could almost be said to be more important than that of sub-aerials, since, as it is known, the large pile of layers accessible to the geologists' investigation,

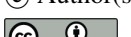



and which also constitute the soil of our peninsula, is almost entirely of submarine origin". (Mercalli 1883)


At the same time, he supposed that accounts for eruptions occurred in the immense solitude of the seas must be sporadic. The 1846 eruption (of which we refer later in this paper), perhaps would have remained unknown if a captain of a merchant vessel had not accidentally been a spectator.

When we refer to sea volcanism, we refer to the volcanism in the Mediterranean Sea. Even if Bermuda Islands have a volcanic origin, volcanism there is not active. While in the Sicily Chanel and all around the coast of Southern Italy, volcanic phenomena are frequent also today. Just to mention the most recent, in November 2002 there was an important degassing event at sea out of the coast of Panarea Island (Capaccioni et al. 2005)


### 3.5 Tempests, storms and sea eruptions

Before going further, we spend a paragraph to clarify the difference between a storm (or tempest), in the specific case sea storms, and eruptions, in the specific case sea eruptions. The word "tempest" is defined
in dictionaries as a violent storm, with high winds that can be accompanied by rain, hail or snow. It is a word with Latin origin, and its etymology indicates an evolution from "period of time" to "period of weather" to "bad weather" to "storm". The word evolved to include also a figurative sense of "violent commotion". WS uses the word "tempest" in this double meaning, when in the play the tempest is also the turmoil that invest the characters' life, maintaining them in a state of temporary and disarming
confusion.

In the first scene of the play we assist to a storm, or at least so it seems. Immediately after, already in the second scene of the play we learn that the storm is the product of Prospero's potent art, and we assist to a second description of it through the words of Ariel reporting how he caused the calamity following
Prospero's directions. Another description comes in this occasion by Miranda witnessing the





shipwrecking from the shore (descriptions analysed in par. 5.2 and 5.3). The initial fatality is reiterated in the words of some characters, during the play.

What happen to the sea during a storm? Its surface is strongly affected by the wind force, in this case

waves are created by the friction between wind and surface waters, and are said wind-driven waves or surface waves. In other words, surface waves are the product of the interaction between the sea and the atmosphere of our planet.

The first scene of the play describes the storm with the words typical of a storm: thunder and lightning.

The boatswain uses words typical of a windstorm such as "blow", "wind", "storm". The word "fire" in this description do not appear neither once (I.i.)

Other hazardous waves can be the result of an underwater disturbances that displace large amount of waters quickly such as earthquakes, landslides or volcanic eruptions. This type of waves can also cause

a tsunamis. The main difference with the previous types of waves described is that in that case only the water surface is interested by force-driven wind. In the second case the great amount of energy released from within the Earth travels up to the surface, displacing water and raising it above the normal sea level.

In particular, during a submarine eruption we assist, as for sub-aerial volcanoes, to the rising of super-heated molten rock (magma) along with ashes and gases. What happen on the surface of the sea depends on how the water and the magma interact, in relation to the depth of the volcano seabed (Németh & Kòsic 2020). In shallow waters, where water depths is less than 100 meters, hydro-volcanic explosion can be violent, but increasing water depth significantly decreases the explosive energy of the

eruptions as the expansion of steam becomes limited (Zimanowski, B. et al. 2003; Clague, D.A. et al. 2000).





A volcanic eruption produces earthquakes, since the magma exerts pressure on the rocks until it cracks the rock. As we will see later (par. 5.2), in *The Tempest* the descriptions of the initial fatality in the following verses of the play is fairly different from the initial scene and correspond better to a sea eruption. It is not a case, we believe, that Prospero never uses the word "storm". He uses the word "tempest" to emphasize that what we saw in the first scene is the product of his magic. The reasons why the initial scene of the tempest is different from the description given by Ariel in vv. 193-206 I.ii and in that given by Miranda observing the event from the shore in verses I.ii. 3-4 (see par. 5.2 and 5.3) could be only speculated. Maybe putting into scene a sea eruption would have been difficult; or maybe WS wanted to emphasize that the initial catastrophic event was the product of Prospero's magic art, and in doing so took inspiration from another natural event certainly more impressive, even if simply described by words.

## 4 The Island of the Tempest

### 4.1 One, no one, hundred thousands islands

We do not intend to precisely locate the island. We take for granted that in conceiving the island, WS assembled, as in a puzzle, features coming from different real and ideal islands. Instead, we intend to collect all the indications that can help us to analyse all elements in the play that can be connected to a real location in terms of an environmental and geophysical asset. Even if there are geographical indications on how both Prospero and Miranda, and then King Alonso of Naples and his crew reached it, the island is a polyhedral place, both in the philosophical-political sense and in a geographical-environmental sense.

The island is the place where the great Renaissance themes revive: the philosophical utopia, the boundaries of human knowledge, the the dominion of nature. We recognize the world of the great journeys, of the newly discovered lands, of the mysterious islands. This is also the land where WS transfers all the problems inherent in colonialism to the new discovered lands. In the relation between





Prospero and Caliban, the savage and deformed slave, we recognize the relation between England and America, the invaders and the natives, as commentators have pointed out (Knight 1984)

Caliban owes its name to the anagram of the word "Cannibal". WS read the famous essay of Montaigne
in John Florio's version, *Of Cannibals*, the only undisputed source of the play. In this Essay Montaigne compares "cannibalism" of some indigenous population in Brazil to the "barbarianism" of the 16th-century Europe. ( Florio 1893)

### 4.2 The Bermuda Hypothesis

If we think of the philosophical and political asset, as described in the previous paragraph, we can certainly agree that WS used sources to get more acquainted with the so called New World. Literary scholars generally agree that he used the so-called Bermuda Pamphlets as a source of the play, a series of narratives of a wreck occurred during an expedition for the colonization of Virginia. In particular, *A*
*true repertory of the wreck and Redemption of Sir Thomas Gates Knight* (Strachey 1625). The true repertory is a letter Strachey wrote to a never identified woman at the English Court. In the letter Strachey reports the 1609 shipwreck on the uninhabited island of Bermuda of the colonial ship *Sea Venture* which was caught in a hurricane while sailing to Virginia. Despite the ship, that was run aground off the coast of the island, the crew stranded on it for almost a year before completing the
voyage to Virginia.

Some commentators found difficult to accept that WS could have had access to confidential material reporting a wreck near Virginia, when the English government was so intent in organizing expeditions to colonize the new lands. The letter was in fact published many years later in 1625. But if we agree that
WS was in contact with John Florio, a royal language tutor at the Court of James I, we can also accept that WS read the letter (Gerevini 2008). In fact, Florio was very well introduced at Court, and as Gerevini refers, in the period of the *Sea Venture* wreck he delivered to the Earl of Pembroke, on behalf of Thorpe, a book of Healey, *The discovery of the New World*, in addition to the WS Sonnets.





Moreover, Florio knew Richard Hakluyt, a leading Adventurer and a member of the Virginia

Company's counselling. In 1580 Florio translated for Hakluyt Cartier's travels because Cartier's books would have served Hakluyt for his travels in the New World. In 1616 a copy of Strachey's letter was found among Hakluyt's stuff. Eventually Florio read the letter thanks to his relation to Hakluyt. But one person who was fundamental in organizing exploratory voyages to the New World was the Earl of Southampton, one of its patrons. Furthermore, it should not be overlooked that Florio had shown, as

WS, a great passion for the sea . From the dedication to the reader of his dictionary we also learn the great knowledge and competence Florio acquired thanks to the translations he did for those who gravitated to the Court, as we have already mentioned. In the dedication we read:

"I was but one to turne and winde the sailes, to use the oare, to sit at sterne, to prike my carde, to watch

upon the upper decke, boate-swaine, pilot, mate, and master, all office in one, and that in a more unruly, more unweildie, and more room-some vessel, then the biggest hulke on the Thames or burthen-bearing Caracke in Spain, or slave-tiring Gallie in Turkie, and that in a sea more divers, more dangerous, more stormie, and more comfortless then any ocean". (Florio 1598 quoted in Gerevini 2008)

It is undeniable that WS spoke very often about the sea in his plays as an experienced sailor. Mark Twain – reports Gerevini – was remarking this aspect, and "if it is Mark Twain to do so, who certainly was an experienced sailor, we can believe him".

To the aim of the present paper, we are interested in observing the way WS used the Strachey's letter,

and if it is possible, to find elements in it able to re-conduct the island to a specific natural environment. One important aspect to be addressed and that we will consider in details in the last section of the paper is how WS deals with natural hazard in the play.




### 4.2.1 Echoes of the Bermuda in The Tempest

In reading Strachey's letter, we can easily find in the play echoes of the faraway transoceanic lands
inhabited by spirits and devils. The Tempest eventually took the atmosphere created by the collective
imagery concerning these lands considered, wrongly according to Strachey (1625), uninhabitable:

"And hereby, also, I hope to deliver the world from a foul and general error, it being counted of most
that they can be no habitation for men, but rather given over to devils and wicked spirits; whereas
indeed we find them now by experience to be as habitable and commodious as most countries of the
same climate and situation, insomuch as if the entrance into them were as easy as the place itself is
contenting, it had long ere this been inhabited as well as other islands". (Strachey 1625)

Strachey then describes the island in details, the nature of the soil, that is one and the same. "The mold
dark, red, sandy, dry and uncapable, I believe, of our commodities or fruits". He also writes "there is not
through the whole islands either champaign grounds, valleys or fresh rivers". Then he describes the
flora mainly with palm trees, cedar and prickly pear. And also emphasizes that there were no rivers nor
running springs of fresh water. The only water to be found in the ground is that coming from the rain:

"When we came first we digged and found certain gushings and soft bubblings, which being either in
bottoms or on the side of hanging ground, were only fed with rain water, which nevertheless soon
sinketh into the earth and vanisheth away, or emptieth itself out of sight into the sea, without any
channel above or upon the superficies of the earth". (Strachey 1625)

He finally describes a very rich fauna especially for what concern fishes.

Despite the atmosphere previously described, none of the ecological traits described in Strachey's letter
can be found in Prospero's island, where there is no indication of tropical vegetation. Instead, trees
typical of temperate climates are described, oaks, pines, wild apple trees, kernels, as well as bushes that





produce berries (Brazzelli 2009). Caliban refers to the fertile areas of the island. Speaking to Stefano and Trinculo he says "I'll show thee every fertile inch o' th' island" (II,ii, 148) and also the springs of fresh water "I'll show thee the best springs" (II,ii,169). From his early interaction with Prospero we learn that the island have different type of waters: "fresh spring" and "brine-pits". Caliban has showed him how to distinguish between them obtaining, as a reward, water with blueberries and language

teaching. The multiple references to rivers and ponds, including a "foul lake" (IV, i, 183), and brambles (" briars ") and other thorny bushes, allow us to identify a real ecology of the island. The tree in which Ariel has been imprisoned for 12 years, before being released from Prospero, is a pine. "Line trees" (V, i, 10), which are perhaps lime trees, not tropical trees, protect from the weather, the entrance of the cave - home of Prospero.


We are also able to identify the geology of the island (Fitz 1976). We know that the coast is cut by coves or nooks, since Ariel feels obliged to explain to Prospero in just which nook he chose to hide the ship (Ariel, I.ii.226-29). We know that there are banks, since Ferdinand sits on one to weep (I.ii.389-90) We know from Ariel that the sands are yellow (I.ii.376) We know that there are large rocks with caves

in them, for Caliban lives in one of them (I.ii.389-90) and Stefano hides his stolen liquor in another (II.ii. 137-38). There are streams and ponds, some fresh (I.ii.339; II.ii.164; II.ii.75) and some polluted (IV.i.182).

For what we have until now described, we can for sure asserts that the Island of *The Tempest* it is not a

tropical island. There are not even any palm trees – Fitz emphasizes – the prime requisite for a modern tropical island, although Shakespeare speaks of palm trees in other plays.

**4.3 Placing the island into a Mediterranean context**

As we will see in the last section, the one dedicated to natural hazard in *The Tempest*, we can place the island into a Mediterranean context not only for the intrinsic features of the island itself. But for the time being, we will follow the geographic route indicated in the play.



An important source of *The Tempest* is Virgil's *Aeneid*. A parallel with Aeneas' route in the
Mediterranean is clearly established (see Fig.4 for Aeneas' route). The Court Party follows a route very
similar to that of Aeneas: from Tunis (the Old Carthage), in North Africa, to Naples near Cumae where
Aeneas meets the Sibyl. In an apparently aimless conversation among Gonzalo, Alonzo and Sebastian,
Gonzalo insists on identifying Carthage with Tunis and the other two insist on repeating the name of
Aeneas and Dido (*Adr.* Tunis was never grac'd befor with  such a | Paragon to their Queen *Gon.* Not
since widow Dido's time| *Ant.* Widow! A pox o' that! How came that | Widow in? Widow Dido! *Seb.*
What if he had said "widower Aeneas" too? | Good Lord, how you take it? *Adr.* "Widow Dido" said
you? You make me | Study of that: she was of Carthage, not of | Tunis   *Gon.* This Tunis, sir, was
Carhage  II.i. 71-87). C. Still states that WS accentuated the importance of this reference to Dido with
the ignorance of the disputants (Still 1921). While the question of Antonio "How come that widow in?"
draws the attention to the parallel between the experience of the Court Party and that of Aeneas in Book
IV of the *Aeneid* (Kott and Miedzyrzecka 1977)

As from the first scene of the play,  a  storm surprises the Court Party while they are navigating from
Tunis to Naples. As we can see in the map (Fig.4), the first trait of sea is the Sicily Channel. But in the
present paper we will refer to the Sicilian Sea not only for the part of the sea in between Africa and
Sicily (including the Sicily Channel, the Malta Channel and the Pantelleria Channel); We will refer also
to the part of the Tyrrenian out of the North of Sicily, where a series of small islands are located (Egadi
and Aeolians).

The hazard of the Sicilian Sea has been known since ancient times. Biblical accounts and historical
sources and the Epic tell us about shipwrecking in this area of the Mediterranean Sea.  In his journey to
Italy, Aeneas was advised by Eleno of Burtroto to avoid Scylla and Charybdis. He did so, and after
having been near the Etna, Aeneas reached Eryx (near Trapani) where his father Anchise died. He, then,
decided to go back to Carthage. During the trip, Juno, who hated the Trojans provoked a tempest
against the fleet. (*Aeneid* I, 81-222).



In the *Act of the Apostles*, Paul of Tarsus, persecuted by the Jews and imprisoned, asked, as a Roman citizen, to be tried in Rome. Under the Governor Porcius Festus, he was sent in Rome by sea. His boat shipwrecked and  the crew reached Malta (The Acts of the Apostles 27; 28 1-10). Echoes of this

shipwreck can be found in Prospero's account on the tempest to Miranda "No, not so much perdition as an hair" (I.ii.30), repeated then by Ariel reporting to Prospero "Not a hair perish'd" (I.ii 219)

Archaeological and more recent remains found in the deep sea testify of a difficult navigation in dangerous water till present times. This is probably due to the complex geodynamic of Italy and the sea

surrounding the peninsula, resulting from the evolution of the borders in between the African and Euro-Asiatic plates. In particular, the Sicily Channel is an area of particular interest in the Mediterranean from a geodynamic point of view (Corti et al. 2006), with a NW-SE oriented  compression and NE-SW oriented extension. Tectonic extension  led  to  an intra-plate rift system characterized by three tectonic faults and a number of underwater edifices, evidence of complex volcanic-tectonic

phenomena (Civile  et  al. 2015). Volcanic episodes occurred in the Sicily Channel in 1831, with the emersion of an island, the Ferdinandea, in the Grahm bank, has led to a monitoring since 1883 by the Italian Navy Hydrographic Institute. Recent hydrographic campaigns have located the most superficial point of the old volcanic building at a depth of about 9 m. This is a potential hazard for vessels (Sinapi et al. 2017)


How many islands populate the sea out the coast of Sicily? Today a list of 105 islands (including major islands, islets, rocks and stacks) is recognized by the Sicilian Island Award (S.I.A.) as valid islands which as a whole constitute approximately 1.11% of all the regional surface (about 285.4 km 2 on a total of 25,711 km 2). For the most part they are rocks or islets, generally of scarce naturalistic interest

and perimetral to the major islands (Muscarella & Baragona 2017). This number is a clear indicator of the complex geodynamic of the area. To this number we have to add the above mentioned Ferdinandea, that emerged from the sea in historical times. The underwater island is located in between Pantelleria and Sciacca (near Agrigento) (see Fig. 4).




Following Aeneas' route (that is supposed to be similar to that of King Alsonso and his crew), some
       commentators identify the islands of *The Tempest* with Pantelleria. Others, crossing the itinerary of the
       Court Party with the events of Sycorax who arrived on the island from Algiers, search the island  along
       the North Africa coasts, identifying it with the island of Lampedusa or Malta (Kott 1974).

More recently, R.P. Roe has advanced the hypothesis that the island is in the Tyrrhenian Sea in the
       Aeolian archipelago. Roe even pointed the island of Vulcano as the possible location of the play  (Roe,
       2011). His arguments agree well with the frame of the present study that sees in *The Tempest* a possible
       source of historical volcanism in the Sicilian Sea. Roe makes a parallel between Aeneas and Alonso's
       shipwreck saying that WS knew exactly where Virgil conceived the lines: "It was a spot where sudden,
violent storms where notorious through history: the sea between the island of Aeolus and the coast of
       Sicily". He then states that WS supplant the roles of Juno and Aeolus with those of Prospero and Ariel.

       Following Roe's directions, it is possible that WS could have taken lot of inspiration from the Vulcano
       island. Roe sustains that the island, as we have already remarked for the Bermuda, was never deemed
permanently inhabitable until the dawn of the twentieth century. Vulcano, as Stromboli, possesses an
       active volcano. It is the Gran Cratere, or La Fossa di Vulcano, and is especially noxious and deadly.
       That's why no one felt confident enough to live permanently on the island until fairly recently. Roe
       refers to the volcanism of the island.  With respect to *The Tempest*, in particular Roe writes about the
       "hot mud pools" to be found in Porto Levante in Vulcanello, a peninsula to Vulcano. The largest of the
pools is impressive with carbon dioxide and sulphur dioxide effervescing through the muddy mixture of
       mineral sludge. The brownish goo bubbles and steams, and stinks mightily. The allusion to this hot mud
       pool is in IV.i.181-184 where Ariel refer to Prospero how he settled the three, Caliban, Stephano and
       Trinculo organizing a conspiracy against him (*Ari.* … At last I left them |I'th filthy mantled pool beyond
       your cell, |There dancing up to their chins, that the foul lake| O'erstunk their feet.)




Verses emphasized later by the entering of the three in the scene soaked with the waters of "the filthy-mantled pool…the foul lake" and stinking to high heaven (*Trin.* "I do smell a horse-piss; at which my nose is in great indignation" IV.i.199-200)

The setting portrayed through the words of Trinculo, Stephano and Ariel, "foul lake", "horse piss", "filthy pool" – concludes Roe – describe exactly the stinking, bubbling, hot mud pool of Vulcano. But what is astonishing is the way Roe explains Ariel defining mud pool with the word "filthy mantled". At the time of WS Vulcano's hot mud would have been "mantled", that is, covered by a floating crust of dry sulphur, and it would have been covered throughout the entire year. This curious natural

phenomenon occurs when bright yellow particulate of sulphur, drifting down from the crater above, collects on the mud pool's surface. Today this coating of yellow mud is seen only during the mud pool's "off-season" that, is, when the pool surface has remained undisturbed by health seeking tourists. At the time of WS, we can imagine the yellow dust remained untouched on much of the rim and slopes of the Gran Cratere, as it did on the hot mud pool in the playwright's day. It can therefore be concluded that

the "yellow sands" the airy spirit sings about in the play ("come into these yellow sands" I.ii.378-381)), refers to the colour of the sulphur.

Roe identifies further evidences in the flora and in the fauna of Vulcano, giving also an explanation for the word "scamels" remained always mysterious for commentators (*Cal.* …sometimes I'll get the /

young scamels from the rocks II.ii.184-185). "Scamels are migratory marsh and shore birds, sometimes found along the Tyrrhenian Seas of Italy and occasionally on beaches in England, or other northern climes". Caliban mention also the Volcano's berries (*Cal.* …Thou strokedest me, and madest much of me;| Wouldst give me |Water with berries in it. I.ii. 333-334). To Roe are clearly the mulberries, berries which proliferated in the wild of Vulcano when the playwright visited the island. Even today, an area on

Vulcan is referred to locally as la "Contrada del Gelso" (the Mulberry district).

Did WS visit Italy? We leave this question for those performing studies on the real identity of WS. Can we really state that the island of *The Tempest* is Vulcano, as Roe does?  As we have already said at the





beginning of this session, we believe that the play is a great container of suggestions that come from

different places imbued with a great imaginary.

## 5 Natural Hazard in The Tempest

### 5.1 A fire-based play


Despite the title, *The Tempest* is a fire based play. Not only for the mythological asset in which all that is suggested by nature become a place of expiation as in Dante's Hell and in the mythological literature of the past. But also from a geo-environmental point of view. Counting words, in the play, the word "water" and "sea" are repeated fifty times in the text, compared to the thirty-four occurrences of air and

the fifteen occurrences of "earth". The word "fire" has not the same frequency of quotations (eleven times) but it is present in the denotations and connotations of its essence (Marnieri 2013).

While water is deprived of its intrinsic power to wet, wrinkle clothes, to drown people, the fire becomes so powerful and frightening that it infects the light of reason. Only at the end of the play the characters

will recover it. Ariel in the form of fire become faster than "Jove's lightnings". The fire is evoked very often under the form of combustion phenomena. To the question why Prospero is so obsessed with woods, asking Caliban and then Ferdinando to continually bear logs, one may answer "firewood", and this is suggested by Miranda when she comforts the log-bearing prince Ferdinando with a personification: "When this burns |"Twill weep for having wearied you" (III.1.19-20) (Jensen 2016).

This is also supported by G. Egan who refreshingly probes deeper in the question of what Prospero plans to use all the wood for. He lives in a cave, still after twelve years on the island (I.ii. 53-55), why does he not build a house? He is strained on an island against his will, why does he not use the wood to build a boat? (Egan 2006). Prospero's boasting about his violent destruction of oaks, pines, and cedars seems to imply that the island was more forested before he arrived. Egan suggests that the point about

the "recurrent arboreal imagery" in *The Tempest* is that Prospero's main activity since his arrival on the island has been its deforestation.



Other important geo-physical phenomena connected with combustion are the St' Elmo fire and *ignis fatuus* (lit., "foolish fire"). In the scene of the tempest, as reported by Ariel, the words seem to evoke the St Elmo's fire: "I boarded the King's ship; now on the beak| Now in the waist, the deck, in every cabin| I flam'd amazement: sometime I'd divide| And burn in many places; on the topmast |The yards and boresprit, would I flame distinctly| Then meet and join". (I.ii.96-101). The St' Elmo fire is a type of luminous plasma discharge from a pointed object, in fields that carry a high voltage. They are often associated with areas of thunderstorms or volcanic ash activity and are completely harmless. A description of it appears also in the Strachey' s letter:

"Only upon the Thursday night, Sir George Somers, being upon the watch, had an apparition of a little round light, like a faint star, trembling and streaming along with a sparkling blaze, half the height upon the main mast and shooting sometimes from shroud to shroud, 'tempting to settle, as it were, upon any of the four shrouds. And for three or four hours together, or rather more, half the night it kept with us, running sometimes along the main yard to very end and then returning… But upon a sudden, toward the morning watch they lost the sight of it and knew not what way it made". (Strachey 1625)

Strachey himself remarks that this phenomenon is frequent also in the Mediterranean Sea. St. Elmo's fire takes its name from St. Elmo (St. Erasmus, a martyred bishop of Italy, who died in 304). He was adopted by the sailors of the Mediterranean as their patron Saint, and the appearance of the fire was afterwards accepted as proof of his protection. But the phenomenon was familiar to the ancient Greeks, and Pliny mentions it in his *Natural History*. When it appeared as a single flame, it was Helena of Trojan war flame, and an omen of ill luck. As a doubled flame it was called Castor and Pollux, the guardian of sailors among the classical gods and therefore a good sign. Another description of St Elmo's fire appears in Hakluyt's "Voyages". It runs:

"I do remember that in great and boisterous storme of this foul weather, in the night, there came upon the toppe of our maine yarde and maine maste, a certain little light, much like unto the light of a little



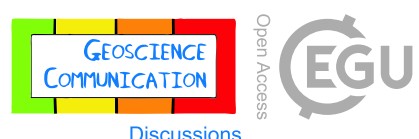

candle, which the Spaniards called the Cuerpo-Santo, and said it was St Elmo, whom they take to be the aduocate of sailers….This light continued aboord our ship abouth three hours, flying from masta to maste, and from top to top; and sometimes it would be in two or three places at once". (Hakluyt quoted by Clark 2005)

The Italian "Corpo Santo" is the origin of the word "corposants", by which the fires were known to English sailors (Clark 2005).

The other interesting phenomenon is the *ignis fatuus* (in the popular culture Jack-o'- lantern or will-o'- the wisp)  "a phosphorescent light seen in the air over marshy places, supposed to be caused by the
evolution and spontaneous combustion of some highly inflammable gas" (Fun and Wagall's New Standard Dictionary – quoted in  Clark 2005). In IV.i. 197-198 Stephano mention it talking about Ariel to Caliban "Monster, your fairy, which you say is a harmeless fairy, has done little better that play the Jack with us". Clark affirms that WS, perhaps, believed that the *ignis fatuus* and St' Elmo's Fire were the same thing, or had a like cause, since he makes Ariel impersonate both phenomena. To us WS was
adding wonder to wonder by exploiting the powerful imagery instilled by the use of fire.

**5.2 A tempest or a sea eruption?**

Till now we have collected evidences that in conceiving *The Tempest* WS was inspired by the
Mediterranean Sea. But where and when exactly the tempest takes place in the play? It is important to answer to understand how the playwright deals with the phenomenon of St. Elmo's Fire. *The Tempest* is the only play where WS respect the time, place and action units of classic drama. The reasons why are not known. Scholars advanced several hypothesis. In any case, we can learn the time in which the tempest takes place, since after having reported on the tempest performed, Prospero ask to Ariel the
time (*Pros*. Ariel, thy charge| Exactly is perform'd: but there is more work. What is the time o'th' day?| Ari. Past the mid season. | The time 'twixt six and now| Must by us both be spent most preciously  I.ii 238-241).



We learn that the tempest occurred surely during the day and not at night. But as we read in the previous
quotes about St Elmo's Fire the phenomenon is described occurring at night. So, what was really
describing WS with Ariel's words:

*"Ari.* I boarded the Kings' ship; now in the beak,

Now in the waist, the deck, in every cabin,

I flamed amazement; sometime I'd divide,

And burn in many places; on the topmast,

The yards and bowsprit, would I flame distinctly

Then meet and join". (I.ii.196-201)

It seems that St Elmo's Fire are described just by the movements of Ariel performing the tempest, and
nothing else. In reading the verses we don't get the impression of Ariel performing "certain little light,
much like unto the light of a little candle" (Hayklut) or of *"*a little round light, like a faint star,
trembling and streaming along with a sparkling blaze" (Strachey). Not at all. On the contrary, in the
description of Ariel we imagine a blazing fire "flaming amazement". The imagery evoked is powerful
and strong and not evanescent as a St. Elmo's Fire. Eventually this phenomenon provoked wonder in
sailors but not in the sense that they could be injured by it.

In reading the whole verses of Ariel's description, we even doubt that WS was describing a tempest:

*"Pros.* Hast Thou, spirit,

Perform'd to point the tempest that I bade thee?

*Ari.* To every article. I boarded the King's ship; now on the beak,

Now in the waist, the deck, in every cabin,

I flam'd amazement: sometime I'd divide,

And burn in many places; on the topmast,





The yards and boresprit, would I flame distinctly,

Then meet and join. *Jove's lightnings, the precursors*

*O'th' dreadful thunder-claps, more momentary*

*And sight-outrunning were not: the fire and cracks*

*Of sulphurous roaring the most mighty Neptune*

*Seem to besiege, and make his bold waves tremble,*

*Yea, his dread trident shake"*. (vv. 193-206 I.ii)

This is not the first time that WS put into scene a tempest, but maybe this is the first time that during a

storm, fire is prevailing while associated with the earth shaking. So, what kind of tempest is this? And
what about the "sulphurous roaring"? In the description, Ariel in the form of fire is faster than Jove's
lightnings.  Neptune (the greek Poseidon), the god of the sea but also the god of earthquakes, is
described intent on shaking his dreaded trident. In Ariel's words there is a war between the sky and the
sea, where the sea is described as the "most mighty Neptune".


As we have already remarked, Neptune was also the god of earthquakes. And he was so powerful to
frighten even Hades lord of the dead, as we can read in the XX book of Homer's *Iliad*

"The sire of gods and men thundered from heaven above, while from beneath Poseidon shook the vast

earth, and bade the high hills tremble. The spurs and crests of many-fountained Ida quaked, as also the
city of the Trojans and the ships of the Achaeans. Hades, king of the realms below, was struck with
fear; he sprang panic-stricken from his throne and cried aloud in terror lest Poseidon, lord of the
earthquake, should crack the ground over his head, and lay bare his moldy mansions to the sight of
mortals and immortals". (Iliad XX. 54-59)


The importance of this deity in the play is also underlined by another circumstance. We have already
said that Caliban owes his name to the anagram of cannibal, representing the good savage of Montaigne
essay *Of Cannibals*. Other claimed that "Caliban" is a word very near to "caribean" to stress the





parallelism between him and the inhabitants of the new colonized lands in Virginia. But Caliban,
beyond being defined by WS "a salvage and deformed slave" in the *dramatis personae*, is very often
described during the play  as half man and half fish (*Trin*. …What| have we here? A man or a fish? dead
or| Alive? A fish: he smell like a fish; a very | Ancient and fish-like smell II.ii. 25-28). And later again:
"Wilt thou tell a a monstruous lie, being but half a fish | And half a monster?" (III.ii.31-33). There is
another Greek god of the sea, Triton, who is the son of Poseidon (Neptune for the romans). Triton is
represented as a merman with the upper body of a human and the tailed lower body of a fish. He was
also depicted as having a conch shell which he would blow like a trumpet. Ovid describes him as "sea-
hued" and with "shoulder barnacled with sea-shell"  (Triton, sea-hued, his shoulders barnacled | With
sea-shell, bade him blow his echoing conch| To bid the rivers, waves and flood retire. Ovid
Metamorphoses 1. 332-335).


As we have already said, this is not the first time that WS uses storms in his plays. As Clark remarks,
thunder, lightning, darkness, and gales are there because they harmonise with the terror, despair, horror,
and wickedness inherent in in his grim plots, and are intended to intensify the dramatic and tragic
atmosphere. We find storms in *Julius Caesar* and *King Lear* two plays belonging to his supreme tragic
period, and in *The Tempest* and *Pericles*, written at the end of his life. Other spells of rough and
tempestuous weather are coincident with dark happenings in *Macbeth* and *Othello*, and in the Winter's
Tale. Often these are "shipwrecking storms", and losses at sea as in *The Tempest*, *Pericles*, *Othello* and
the *Winter's Tale*. (Clark 2005)

The question is now, how WS deals with storms and shipwrecking in the other plays? Do we find in the
other descriptions   words as "fire", or adjectives as sulphurous" or terms connected to earthquakes? In
*Julius Caesar*, in the night before Caesar's assassination Casca and Cicero are having a conversation on
the particularly violent equinoctial storm. And Casca describes the storm using the word "fire" (But
never till to-night, never till now | Did I go through a tempest dropping fire I.iii.9-10). But from the
whole conversations of the different characters on the storm, the word "fire" it is just a way to
emphasize that something terrible is going to happen.



Again in *King Lear*, as in *The Tempest*, associated with a storm, we find words as "fire" and adjective as "sulphurous" ("you sulphurous and though-executing fire" III.2.4), but here the words are used to

connect human tragedy with the scowling front of nature. And to go to the plays where storm is associated with shipwrecking, in *Pericles*, we find again the use of the adjective "sulphurous" associated with storm (Thy nimble sulphurous flashes (III.i.6), and in III.ii 14-15 we find, as in *The Tempest*, storm associated with earthquakes ("Our lodging, standing bleak upon the sea| Shook as the earth did quake"). Nevertheless, in none of these descriptions we find a combination of blazing fires

with the earth shaking, as in *The Tempest*.

Earthquakes are not by chance in this verses since in the final part of the play Prospero in resuming all the prodigies accomplished thanks to his potent art not only he says to have triggered a war between heaven and earth (*Pros*. …And 'twixt the green sea and the azured vault | Set roaring war V.i. 43-44);

he also asserts: " *Pros.* …The strong-based promontory | Have I made shake V.i. 46-47). No doubt, among his many prodigies, Prospero has provoked earthquakes. In a short paragraph dedicate to earthquakes in Shakespeare, Clark affirms that WS had a limited knowledge of earthquakes feeling his own limitation due to a scarce experience of them. Eventually was interested more with the effects of this catastrophic upheavals rather than embark upon an effort to discover their obscure causes. This is

the reasons why references in his plays are few (Clark 2005). Nevertheless, as we have showed, in *The Tempest*, references to earthquakes seems not to be allegorical but more descriptive of natural phenomena.

**5.3 Volcanism in the Sicilian sea and The Tempest**


What the Sea of Sicily has to envy to the Bermuda triangle? Absolutely nothing. Its hazard has been known since ancient times, as we have already seen in historical accounts from the Epics and the Bible. Aeneas and St. Paul shipwrecked near the Sicily Channel that is often in the current chronicles for the route of the migrants approaching Italy from Africa.






Sea volcanism in shallow water not so far from the south of the Sicilian coast (at a distance of 50 km) has been well documented on the occasion of the emerging of the Ferdinandea Island in 1831 (see Fig 6 for the complexity of the seabed in this area). This represents the only well-documented volcanic event occurred in the area; other volcanic activities were uncertainly reported in the surroundings of Graham

Bank during the first Punic war (264-241 BC) (Guidoboni et al., 2002; Bottari et al., 2009), in 1632, 1833, and 1863 (Antonioli et al., 1994; Falzone et al., 2009). Moreover, numerous episodes of strong gas releases in the Graham Bank area were observed in 1816 (Mercalli, 1883), 1845, 1942 and more recently in 2003. (Cavallaro&Coltelli 2019)

As we have already said, a thorough study of how many volcanic episodes occurred in the past in the Sicilian sea has never been performed. We have also reported Mercalli's opinion on the importance of such studies (see. Paragraph 3.4). In 2006, following the directions of Mercalli reporting the18th June 1845 sea eruption episode occurred in proximity of the Grahm bank, a sea expedition, organized by Domenico Macaluso and Giovanni Lanzafame, has revealed a huge undersea volcanic complex, with

more or less the size of Mount Etna. It has been named Empedocle by his founder D. Macaluso and registered at the Royal Geographical Society as Mac06. (Macaluso 2016)

Only recently the seafloor of the Sicilian Sea in proximity of the Sicily Channel is object of in-depth studies which reveal the complexity of the area *(Corti et al. 2006; Cavallaro& Coltelli 2019)*. Moreover,

the evidence of recent eruptions at the Terrible, the historical activity of Ferdinandea and its high-flow fumarolic field, albeit with due caution regarding the volcanism at Banco Nerita, confirm the idea that the submarine relief facing Sciacca is home to volcanic area, active and large (about $25 \times 30$ km); consequently, there is the possibility of a resumption of volcanic activity in an area relatively close to the southern coast of Sicily, within a radius of a few tens of kilometers from Capo San Marco and

Sciacca. (Falautano 2010)




Eventually volcanic eruptions at sea in this area may have occurred also at the age of WS. How it would have appeared to those navigating the area is well described by the already quoted 18[th] June 1845 episode in Mercalli, which we report here in full:


"The 18th June 1845 at about 9.30 p.m., the English vessel Victory being at 36o 44'36" latit. e 13o 44' 36'' longit. (Greenwich?), was violently shaken and its two masts were suddenly overturned as under the effect of a terrible tempest even if in that moment the weather was calm. Suddenly, sulphurous exhalations spread over the air so intense that the crew was almost unable to breath. The vessel was a

bit injured but moved away and from far away the travellers saw three huge fire balls coming up from the sea and the phenomenon was visible for six minutes". (Mercalli 1883)

Surely this description is not so different from the tempest described by Ariel's words previously analysed (vv. 193-206 I.ii) , where a storm occurs, but it is not raining "The sky, *it seems*, would pour

down stinking pitch" says Miranda to Prospero after having witnessed the tempest from the shore (I.ii.3-4); the crew don't get wet (*Ari:* "On their sustaining garments not a blemish, but fresher than before (I. ii. 218-219); where the fire blazes in different places as the three huge fire balls in the description reported in Mercalli (Ari: I'd divide| And burn in many places; on the topmast, | The yards and boresprit, would flame distinctly,| then meet and join Act I.ii 198-201). Moreover, words related to the

wind, as in the first scene, are completely missing.

A further evidence of WS's familiarity with sea volcanism and its effects can be inferred from an older source, *The Geography* of Strabo describing an episode out of the Aeoalian  Island in the Tyrrenian sea. Volcanism in this area is very well studied and even recently in proximity of Basiluzzo, scientists have

discovered what they have called a "smoking land". The name is due to the presence of more than 200 volcanic chimneys, a large number of them are wide and high active, and some of them showed emission of low temperature hydrothermal fluids of marine origin characterized by acidified chemical conditions (Esposito 2018). In the following, Strabo describes sea volcanism  in between Vulcano and Panarea:






"Again, many times flames have been observed running over the surface of the sea round about the islands, when some passage had been opened up from the cavities down in the depth of earth and the fire had forced its way to the outside. Poseidonius says that within his own recollection, one morning at daybreak, about the time of summer solstice, the sea between Hiera (Vulcano) and Euonymus (Panarea)

was seen  raised to an enormous height, and by a sustained blast remained puffed up for a considerable time,  and then subsided; and when those who had the hardihood  to sail up to it saw dead fish driven by the current, and some of the men were stricken ill because of the heath and stench, they took flight;  one of the boats, however approaching more closely lost some of its occupants and barely escaped to Lipara with the rest, who would at times become senseless like epileptics, and then afterwards would recur to

their proper reasoning faculties". (*The Geography* of Strabo 6.2.11)

In *The Tempest*, the event occurring out of sea, besides being described by Ariel to Prospero in the verses already quoted, is also witnessed by Miranda from the shore. She clearly says:

"The sky, it seems, would pour down stinking pitch,
But that the sea, mounting to th' welking's check,
Dashes the fire out" (*Mir*. I.ii. 3-4)

The words recall Strabo's description ("the sea between Hiera (Vulcano) and Euonymus (Panarea) was

seen raised to an enormous height*")*. To Roe, Miranda' s words describe exactly the phenomenon of fumarole erupting (Roe 2011).

In the excerpt quoted, Strabo reports also the effects of gas-inhaling, frequently described in local mythology. The effects of gas-inhaling is reported also in *The Tempest*, when those characters surprised

by the tempest, are described in Ariel's words (*Pros.* My brave spirit! | Who was so firm, so constant, that this coil| Would not infect his reason? *Ari.* Not a soul| But felt a fever of the mad, and play'd| Some tricks of desperation. I.ii.  206-210).





### 5.4 The Tempest in the light of geo-mythology


WS's familiarity with volcanism in the Mediterranean is also supported by the mythology present in the play. Early human civilizations used myths to organize and convey information for transmitting the wisdom necessary to live in harmony with and survive in nature (Lanza & Negrete 2007). In particular, geo-mythology helps to find in the myths the link with geological and natural phenomena that has

generated them. The term 'geo-mythology' indicates a new discipline based on the idea that some myths and legends can be explained in terms of actual geological events witnessed by various groups of people. The term was originally conceived as the geological application of the term 'euhemerism', from the Sicilian philosopher, Euhemerus (300 B.C.), who held the belief that the gods of mythology were simply deified mortals (Vitaliano 1973). In this sense, we may consider WS a further witness of

volcanic phenomena taking place in the Mediterranean.

The already quoted gas-inhaling is present in Mediterranean mythology describing the activities of the Pythia, the priestess at Delphi in Greece. Geologists and toxicologists have published their data suggesting that the trance-like state of the priestess, the oracle at Delphi, was not just a piece of fantasy

(Piccardi 2000, Spiller et al. 2002).

One may think that also the Sybil in Cumae where Aeneas stopped over for prophecies during his trip to Rome may have prophesied under the effect of gas-inhaling. The philosophers of the time (Sophocle, Strabo, Virgil) reports of an oracle of the dead in the Phlegraean Fields, near Lake Avernus (around

Naples, Italy), a very active volcanic area with sulphur vents and boiling springs. The Sibyl, a prophetess, was considered the bridge between the living and the dead. The places evoked in *The Tempest* thorough the route of the Aeneas/Court party are very often associated with hell because of the volcanism. The Averno lake was considered one of the passage to hell as the Etna. At the time of WS there was a legend about the Queen, according to which since Elizabeth I made an agreement with the

devil to be sustained during her reign, when she died (in 1603) was taken by the devils and thrown





inside Etna. Eventually Sicily was considered for its volcanism the land of the devils, not less than the Bermuda Islands (*Ari* …the King's son, Ferdinand, |With hair up-staring, then like reeds, not hair, - | Was the first man that leap'd: cried, "hell is empty, |And all the devils are here"  I.ii213-215)

In Act IV.I to celebrate the marriage between Ferdinando and Miranda, Prospero put into scene a masque where the protagonist is Ceres (the Greek Demeter) whose daughter was raped by Pluto, the king of the hell.  Was this myth imported by Greece or was it conceived directly in Sicily? Classical sources as Diodoro Siculo, Cicero, Ovid places the myth in Enna in Sicily. Near Enna there is a lake, Pergusa, that is believed to be the place where Pluto raped Proserpina.


"Not far from Henna's walls there is a lake, Pergus by name, its waters deep and still; it hears the music of the choiring swans as sweet as on Caystros' gliding stream. Woods crown the waters, ringing every side, their leaves like awnings barring the sun's beams. The boughs give cooling shade, the watered grass is gay with spangled flowers of every hue, and always it is spring. Here Proserpina [Persephone]

was playing in a glade and picking flowers, pansies and lilies, with a child's delight, filling her basket and her lap to gather more than the other girls, when, in a trice, Dis [Haides] saw her, loved her, carried her away--love leapt in such a hurry*!" (*Ovid, Metamorphoses 5. 462*)*.

This myth is associated with the idea of death and rebirth, as the succession of the seasons but also with

the destruction provoked by volcanic eruptions and the following florid re-birth. The most important element associated with Demeter, who was the Goddess of Fertility was the grain. The grain and the volcanoes are the two most important elements associated with ancient Sicily. In ancient mythology Demeter contended the island with Efesto the god of volcanoes. In that occasion the nymph Etna (who gave the name to the most important Sicilian volcano) was the intermediate.


Finally, Ariel disguised as an Harpy interrupt the scene of the banquet (III.iii). Aeneas meets harpies in the Strofadi islands in Greece. Virgil put these figures in the lobby of the hell.





## Conclusion

In the present paper we have collected evidences that some verses of *The Tempest* rather than describing a storm describes phenomena of volcanic origin. WS could therefore have been inspired by accounts and sources describing Sicily and the Sicilian sea. Its hazard has been known since ancient times. Eventually the sea was renown for the "strange things" happening into it: balls of fire, sulphurous exhalations, dead fishes and violent storms occurring when the weather was calm. How amazing should have appeared all these phenomena to the people navigating these waters! Especially in the past when the study of volcanoes was moving its first steps. WS who used natural phenomena to intensify the most dramatic moments of his plot, knew those seas that the sailors dreaded. We don't know if WS ever visited these places. As we have reviewed, he used ancient sources as Virgil's *Aeneid*, the *Bible*, *The Geography* of Strabo. But he may as well have used yet unknown sources as board diary of the vessels navigating those seas. His last play seems really to be a portrait of Sicily and the sea surrounding the island: the land of volcanoes, the land of the tempests of fire, the land of the devils!

## Acknowledgments

This paper is dedicated the memory of prof. Enzo Boschi who recently passed away. He was a scientist but also a passionate of literature and the Arts. Besides making the history of Seismology in our country, he made an important contribution to the development of historical seismology in Italy. The paper is also dedicated to the memory of Dr. Alberto Gabriele of the International Centre for Scientific Culture in Erice, Italy. It was there that for the first time I saw a picture of the Ferdinandea island that gave me the idea for the present study. They were both fan of the island.

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





**Figures**

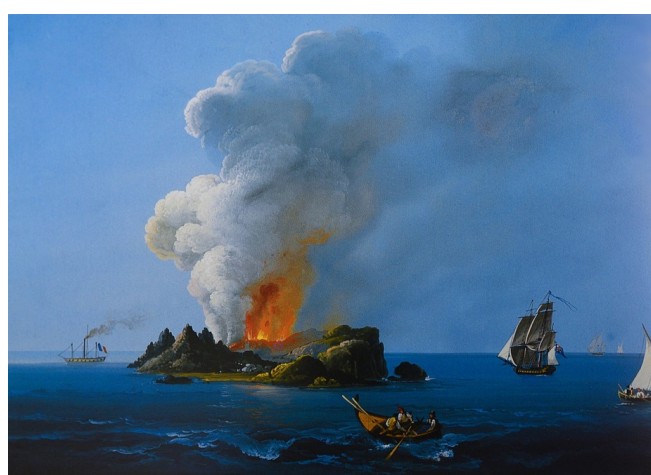


**Figure 1: The Fedinandea island in a painting by Camillo de Vito 1831 source Wikipedia**

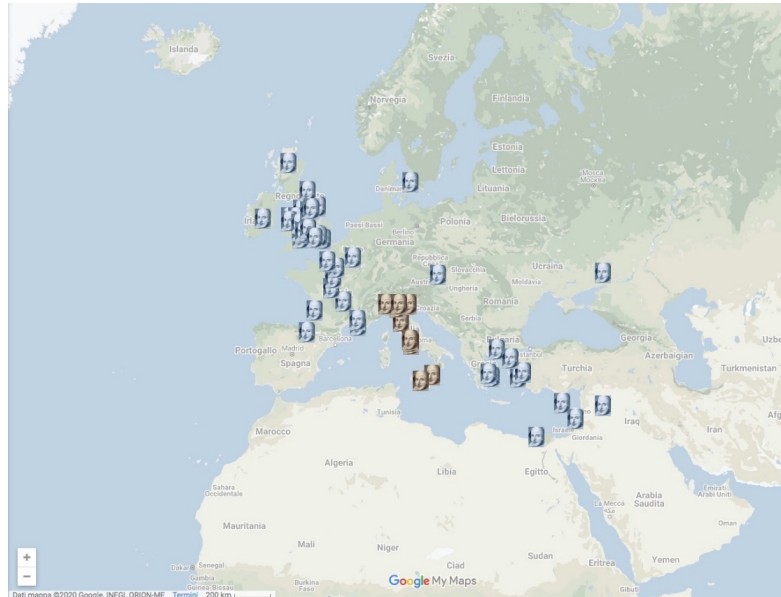





**Figure 2: Map generated with Google Map adding Shakespeare's Locations**

**The icons in brownish colour indicate the plays located in Italy. © 2020 Google**

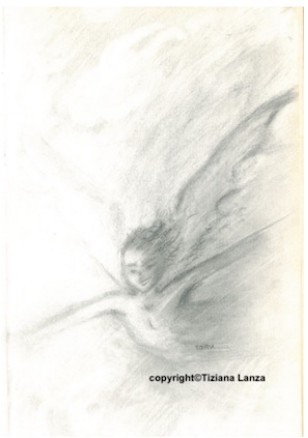


**Figure 3: Ariel, drawing by T. Lanza copyright ©Tiziana Lanza**


**Figure 4: The route of Aeneas in the Mediterranean sea,**
**© Associazione Rotta di Enea http://www.rottadienea.it**



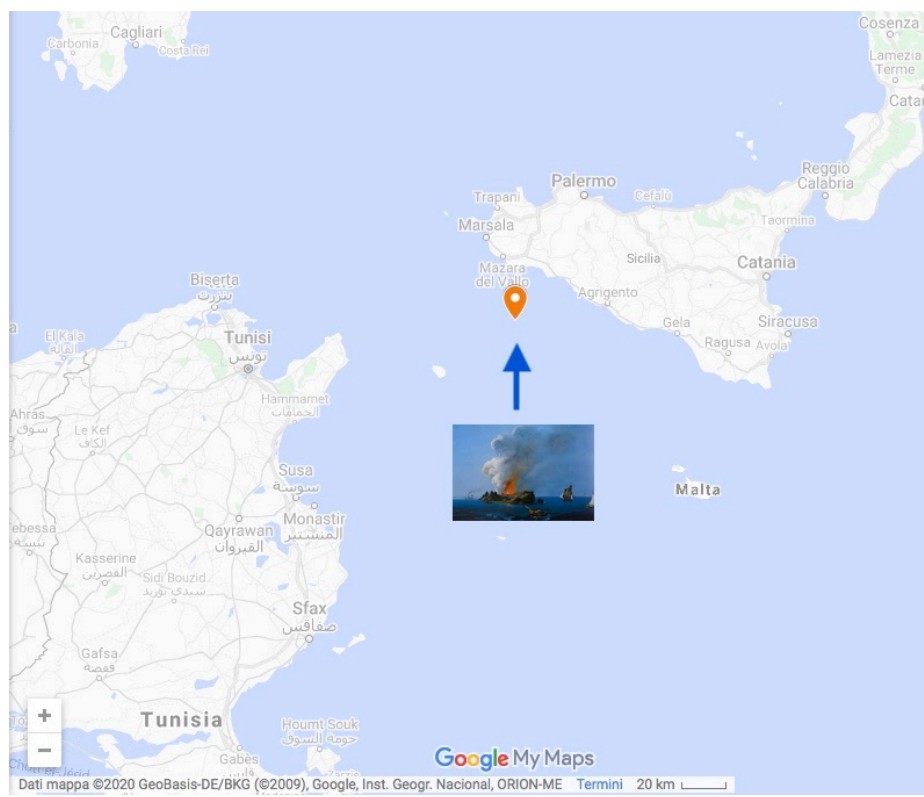

**Fig. 5 Location of the Ferdinandea Island. The island is at just 8 m (26 ft) below sea level and about 50 km from the shore. The map was generated by the author using google maps. ⓒ 2020 Google**



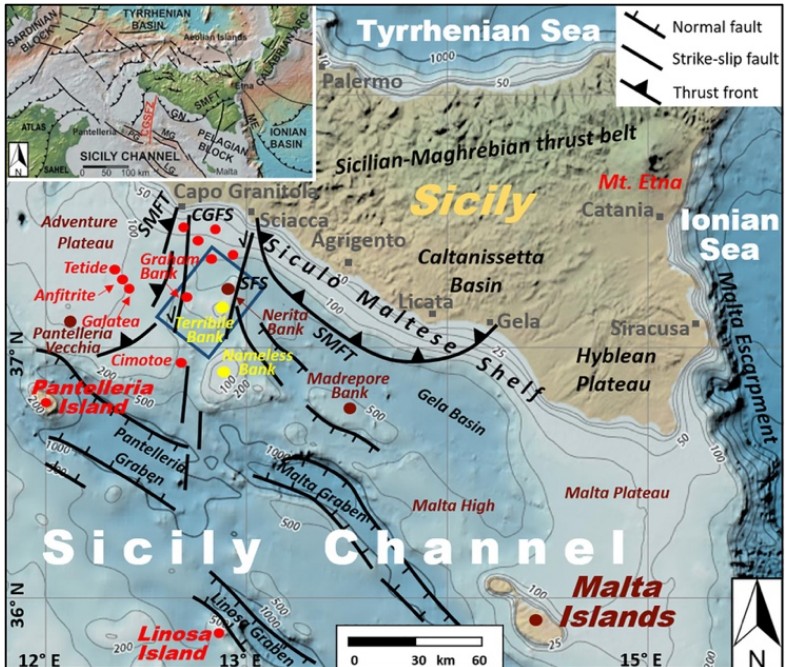

**Fig.6 Shaded-relief bathymetric map of the northern portion of the Sicily Channel (from GEBCO-General Bathymetric Chart of the Oceans-Digital Atlas).; the red, brown and yellow circles indicate the location of volcanic centers, sedimentary banks and sedimentary banks with scattered volcanic manifestations on top. (From Cavallaro& Coltelli 2019)**


