# Peer review of "Using geosciences and mythology to locate Prospero's Island"

_Geoscience Communication, 2020_

## Referee Comment (RC1) · R. John Leigh (Referee) · 13 Aug 2020

Review of: Using geosciences and mythology to locate Prospero's Island GENERAL COMMENTS: Tiziana Lanza has written an informative and interesting review of the sources – both scientific and literary – that William Shakespeare (WS) may have used in writing The Tempest. Her account is balanced, appropriately weighing the evidence for each suggestion, and well constructed. Her writing style will make this article accessible and appealing to a wide readership. My criticisms are relatively minor, concerning a few points of evidence, and making suggestions for improvements in English style (although the writing is generally very good). One minor criticism is that the paper is sometimes a bit repetitive (for example, the emergence of the Ferdinandea Island is referred to several times). The author could easily address this. SPECIFIC

COMMENTS AND TECHNICAL CORRECTIONS: (please treat style issues as suggestions): Abstract, line 8: "...in the context of naturally occurring hazards." (hazards – plural) Abstract, line 12: "..proposed several hypotheses..." Abstract, lines 14-15. "We don't intend to identify the island, we believe the island is a polyhedral place both from a philosophical-political and from a geographical point of view." This sentence might throw some readers. Although "polyhedral" (many sided) is correct, some readers might think that this refers to the island itself having many sides. Perhaps something along the lines: "Our goal is not to identify the island but rather to examine the various geographical and philosophical-political factors that may have influenced Shakespeare's literary creation." Abstract, line 19: "phenomena originating in the volcanism..." Abstract, line 20: I do not understand what "sources precious" means. In the text, Strabo is referred to. Perhaps "historical sources" is meant? Line 22: delete "In recent years" – rather, "A new trend towards..." Line 31: "for calculating, for instance, the return period of an earthquake" – suggest "for predicting future earthquakes." (If this is technically correct). Line 36: "Details about the new updated front-end..." This is jargon. Please state simply what is meant. Are these details of predicted events? Line 40: "...since many years..." Better as "...for many years..." Line 54: "While in.." better as "During..." Line 63: "only a couple of centuries later" better as "over two centuries later" (if correct). Line 69: "wordily renown" better as "worldly renowned". Line 71: "In doing so, we intend to do a work..." better as "In doing so, we intend to conduct a work..." Line 97: Comment: "some of the neo-scientists" who were tolerant of alchemy included Newton. Lines 102-103: "Nevertheless we assist to a slow decline..." The meaning is unclear. Was this the intended meaning – "Nevertheless, we can observe a slow decline in the importance of the Renaissance Magus."? Line 109: Comment - Perhaps add mention of Galileo, who showed the importance of experiments (and who used mathematics more than Bacon). Lines 111-112: The meaning of the following sentence is not entirely clear: "Nevertheless, Bacon himself reinterpreted one of the main features of the magical-alchemical tradition: the philosophy of domain, a knowledge aiming at transforming nature (Rossi, 1968)" Not everyone

will want to read Rossi's book to clarify the sentence. Please make this clearer. Do you mean that Bacon was saying that alchemists had a different set of values? For example, in his Aphorisms, Bacon wrote: "It is true that alchemists have some achievements from their labors, but these came by chance, incidentally, or by some variation of experiments, such as mechanics are accustomed to make, and not from any art or theory; for the theory they have formed brings more confusion than help to their experiments." Line 115: better as "Nevertheless in this respect. . ." (Also, an alternative explanation to the one mentioned here is that WS knew Bacon and discussed scientific matters with him.) Line 135: reads better – "Prospero uses the adjective "rough" when referring to his magic. Also, commentators remark. . ." Lines 154-163: Comment: Although I agree that more attention should be given to John Florio as a candidate for the authorship of Shakespeare's works, perhaps this section should be put in a more neutral voice (your enthusiasm may turn off readers who are not ready to entertain this possibility, such as Stratfordians, who are the majority of readers). Line 194-5: ". . .usurped his title of the Duke. . ." Line 203: Perhaps: "Alonsa, and his son Ferdinando, are searching for each other." Line 210: Perhaps: ". . .final celebration clouded by Prospero's unexpected melancholia." Line 218: Having stated that the sources of the play are debated, the sentence starting "He used sources. . ." should better start: "He appears to have used sources to build the philosophical and political assets of the play such as, for instance. . ." (or similar to imply that these are hypotheses). Lines 253 and following. The general consensus would seem to be that Stachey's description is most consistent with a hurricane – which frequently affect that area of the Atlantic. Perhaps state that. Line 285: ". . .eruption we assist.." is unclear. Assist does not seem to be the right word. Perhaps "observe"? Line 314: Is "multifaceted" better here than "polyhedral"? Line 325: Better to say "It is generally agreed that WS read the famous. . ." Line 345: WS might also have seen the Strachey letter if he was in contact with a member of the Second Virginia company. Line 353: ". . .Hakluyt's belongings." (reads better) Line 373: "natural hazards.." (plural better) Line 408: "..learn that the island has different. . ." Line 424: "..for sure assert that. . ." Line 430: This sentence could be improved. Perhaps "As

we will see in the last section, which is dedicated to natural hazards in The Tempest, there are other reasons besides the island's intrinsic features to place it in the Mediterranean. But for the..." Lines 439-443: Consider breaking this quoted conversation out into separate lines for each speaker – so it can be better appreciated. Line 449: Should "trait" be "tract"? Is trait the correct technical term here? Line 453: "..to the part of the Tyrrenian lying to the north of Sicily.." Line 456: specify ""...and Virgil's Epic.." Line 468: "...testify to a difficult..." Line 469: Should "geodynamic" be plural (here and elsewhere in the paper)? Line 484: km2 (superscript "2") Line 485: Is "perimetral" a technical term? Do you mean outside of the perimeter of the major islands? Line 501: "...WS supplanted..." Line 503: "Following Roe's directions..." Perhaps, "Consistent with Roe's suggestion..." Line 504: "Roe maintains.." Line 538: "To Roe, these are clearly..." Line 545: "...with at great imagination." Line 619: "..collected evidence..." Line 622: "..WS respects.." Also, I do not really understand what this sentence about WS respecting the time, place, and action units of classic drama means; please clarify. Line 624: Do you mean that the tempest that WS uses as a source must have occurred before the performance date of the play in 1611? If so, please clarify. Line 640: "...Fire is described..." Line 714: "...and thought-executing..." (thought) Line 728: "Eventually, WS was interested..." Line 729: "...rather than embarking upon..." Line 736: Suggest: "What has the Sea of Sicily to envy in the Bermuda triangle?" Line 758: "...Channel has been an object of..." Line 767: I am not clear what this sentence starting "Eventually volcanic eruptions at sea..." means. Do you mean that it was probably known at the time of WS that sea eruptions occurred? Line 771: Make sure to superscript the degree symbol, such as 36o Line 850: Can you cite a source for this legend about Elizabeth I being thrown inside Aetna? Line 875: "...Harpy interrupts the..." Line 882: "...collected evidence that..." Line 902: "...both fans of..." REFERENCES: Please make sure that the references are in alphabetic order (if this is the required format). At present, Strachey is followed by Jensen. Also, in the opening parts of the paper, state that you will be using the Arden 1986 edition as the source for your quotes from The Tempest. FIGURES: Figure 2: Who generated this figure using

Google Map – the author? If not provide the citation. Is using the face of WS the best symbol? Perhaps use colored dots or crosses to avoid overlap? Figure 3 is charming, but is it necessary in this paper? Figure 4 is referred in the text in line 488. Should this be a reference to Figure 5?

---

## Referee Comment (RC2) · Anonymous Referee #2 · 3 Sep 2020

The topic broached by the author is a stimulating one, and I feel a genuine interest in the argument developed by the author. However, to my mind, the paper should be fully rewritten to be convincing and to appeal not only to specialists of geoscience, but also to Shakespeareans themselves. This is worth it. The first thing the author needs to consider is the length of her paper. It is much too long, all the more so as the first 12 pages or so seem more or less irrelevant and do not probe the issue of volcanism in the play. Generalities should be removed, as well as confusing considerations on Shakespeare's authorship (Bacon, Florion), which have nothing to do with the scientific argument put forward here.

De facto, it contains too many factual errors and inaccuracies, especially regarding Shakespeare and the play itself. For instance, -The Tempest is NOT "considered the

last play of WS mainly because it is a container of all the themes" previously dealt with by the playwright. -The Tempest is NOT "the only play where WS respect the time, place and action units of classic drama". -Shakespeare's favorite setting is NOT the UK. -Alonso's son is NOT named Ferdinando. -The first storm may indeed be an illusion, which explains why the shipwrecked victims remain dry. But there are several storms on the island, and they are all treated differently (see Jones and Chiari for example).

Surprisingly, while the paper takes stock of recent trends in Shakespeare criticism, some fairly recent studies on ecocriticism (on Bacon in connection with The Tempest–see Popelard–and on storms, particularly) are never mentioned in this study, whereas obsolete works are repeatedly quoted. As a result, the bibliography should be updated with good references (Dan Falk, for example, cannot be said to be an absolute reference).

Besides, the English should be checked and corrected by a native speaker: many sentences are awkward and the use of grammar is not always correct. For example: -[In recent years] a new trend towards the re-unification of the two main streams of culture, the humanistic and the scientific, is becoming more evident year by year. -Ovid's Metamorphosis (we should read Metamorphoses) -Archaeological and more recent remains found in the deep sea testify of a difficult navigation in dangerous water till present times. (we should read testify to) These mistakes are numerous and prevent a smooth reading of the text as a whole.

Methodologically, the author never relies on early modern translations, which is a problem. For we know that early modern translations were imperfect, and that there were important variants in the translated texts. Moreover, which texts were translated in Shakespeare's time, and which were not? What access did he have to Strabo, for example? Regarding mythology, the same problems crop up. Neptune was the god of earthquakes, but he was not particularly known as such in Shakespeare's times. It would therefore be crucial to study early modern representations rather than foregrounding our own perceptions of mythology.

Finally, some premises seem particularly frail. Shakespeare probably knew sailors, yes. But how can that be proved? How do we know what he learnt from their testimonies? More importantly, what did we know of the Vulcano islands (since Prospero's island would be partly inspired from this specific location), in the period? If Shakespeare knew about vulcanism, why does the author never quote any early modern text devoted to this particular phenomenon?

In the play text, words such as sulphur and fire are present, but that is not enough to assert that Shakespeare describes a volcanic phenomenon (sulphur, as a matter of fact, related to hell in the early modern period, and it had also much to do with the pyrotechnics used for the stage–and that could have been a challenging argument). He may very well describe, as has been argued elsewhere, a hurricane (and no need for that to rely on a wind imagery–suggestive images and evocative sentences were the very essence of early modern drama). The paper actually heavily relies on Roe's arguments, but these have not always been regarded as convincing by Shakespeare critics: the analysis of the text remains too superficial to be enlightening.

---

## Referee Comment (RC3) · R. John Leigh (Referee) · 25 Sep 2020

The author has responded well to my criticisms. I look forward to seeing the final ms. It is an interesting paper that will appeal to a broad readership. John Leigh
* * *

---

## Author Comment (AC1) · 25 Sep 2020

Thank you R. John Leigh

for taking the time to review my paper, and in particular I am grateful for your specific comments and technical corrections that will surely help improve it.

I herewith address my line-by-line comments:

One minor criticism is that the paper is sometimes a bit repetitive (for example, the emergence of the Ferdinandea Island is referred to several times). The author could easily address this.

Yes, I will shorten the paper to make space also to other relevant issues not considered

in the present draft.

Line 8: will be changed in "in the context of naturally occurring hazards"

Line 12 "Literary scholars have done several hypotheses" will be correct in "Literary scholars proposed several hypotheses"

Line 14-15 will be changed in "Our goal is not to identify the island but rather to examine the various geographical and philosophical-political factors that may have influenced Shakespeare's literary creation."

Line 19 will be changed in "phenomena originating in the volcanism"

Line 20: I do not understand what "sources precious" means. In the text, Strabo is referred to. Perhaps "historical sources" is meant?

I refer here to what I discussed in par. 1 , lines 58-64. Line 20 can be rewritten in this way "could have used historical sources still unknown and precious. . ."

Line 22 "In recent years" will be deleted

Line 31: "for calculating, for instance, the return period of an earthquake" – suggest "for predicting future earthquakes." (If this is technically correct).

"Return period" is technically correct and it is a different concept with respect to earthquake prediction.

Line 36: "Details about the new updated front-end. . ." This is jargon. Please state simply what is meant. Are these details of predicted events?

"front-end" it is here referred to the data-set on past-earthquake (this is what the catalogue is) see here: http://storing.ingv.it/cfti/cfti5/. The data have been collected to compile a data base of past earthquakes also using historical sources. But I will change the sentence to make it more clear.

Line 40 "since many years" will be correct in "for many years".

Line 54: "While in.." will be correct in "During. . ."

Line 63: "only a couple of centuries later" will be correct in " two centuries later"

Line 69: "wordily renown" will be correct in "worldly renowned".

Line 71: "In doing so, we intend to do a work..." will be changed in "In doing so, we intend to conduct a work. . ."

Line 97: Comment: "some of the neo-scientists" who were tolerant of alchemy included Newton.

Yes, but here I refer to the Elizabethan Age.

Lines 102-103: "Nevertheless we assist to a slow decline. . will be changed in: "Nevertheless, we can observe a slow decline in the importance of the Renaissance Magus."

Line 109: Comment - Perhaps add mention of Galileo, who showed the importance of experiments (and who used mathematics more than Bacon)

I will add mention of Galileo.

Lines 111-112: The meaning of the following sentence is not entirely clear: "Nevertheless, Bacon himself reinterpreted one of the main features of the magical-alchemical tradition: the philosophy of domain, a knowledge aiming at transforming nature (Rossi, 1968)"

It will require more than a couple of lines to explain clearly the concept, so I prefer to delete the sentence.

Line 115 I will reconsider the question of Shakespeare's authorship. I will mention it once and for all at the beginning of the article.

Line 135: will be changed in "Prospero uses the adjective "rough" when referring to his magic. Also, commentators remark. . ."

Lines 154-163: Comment: Although I agree that more attention should be given to John

Florio as a candidate for the authorship of Shakespeare's works, perhaps this section should be put in a more neutral voice (your enthusiasm may turn off readers who are not ready to entertain this possibility, such as Stratfordians, who are the majority of readers).

The same for Line 115.

Line 194-5: will be changed in ". . .usurped his title of the Duke. . ."

Line 203: will be changed in "Alonso, and his son Ferdinando, are searching for each other."

Line 210: will be changed in "final celebration clouded by Prospero's unexpected melancholia."

Line 218: Having stated that the sources of the play are debated, the sentence starting "He used sources..." should better start: "He appears to have used sources to build the philosophical and political assets of the play such as, for instance. . ." (or similar to imply that these are hypotheses)

Will be changed according to your suggestion.

Lines 253 and following. The general consensus would seem to be that Strachey's description is most consistent with a hurricane – which frequently affect that area of the Atlantic. Perhaps state that.

Yes, this could be stated in the par. dedicated to the Bermuda Hypothesis.

Line 285: "...eruption we assist.." is unclear. Assist does not seem to be the right word. Perhaps "observe"?

"assist" will be changed in "observe"

Line 314: "polyhedral" will be changed in "multifaceted"

Line 325: will be changed in "It is generally agreed that WS read the famous. . ."

Line 345: WS might also have seen the Strachey letter if he was in contact with a member of the Second Virginia company

Yes. . .I will reconsider all this.

Line 353: will be changed in ". . .Hakluyt's belongings."

Line 373: will be changed in "natural hazards.."

Line 408: will be changed in "..learn that the island has different. . ."

Line 424: will be changed in "..for sure assert that. . ."

Line 430: Will be changed in "As we will see in the last section, which is dedicated to natural hazards in The Tempest, there are other reasons besides the island's intrinsic features to place it in the Mediterranean".

Lines 439-443: Consider breaking this quoted conversation out into separate lines for each speaker – so it can be better appreciated.

I will do it, for all the quoted long conversations in the paper.

Line 449: "trait" will be corrected in "tract"

Line 453: will be changed in " to the part of the Tyrrhenian lying to the north of Sicily "

Line 456: will be changed in " Virgil's Epic.."

Line 468: will be changed in "testify to a difficult"

Line 469: "geodynamic" will be changed in "geodynamics" all over the paper.

Line 484: km2 (superscript "2") will be corrected accordingly.

Line 485: Is "perimetral" a technical term? Do you mean outside of the perimeter of the major islands?

I report Muscarella and Baragona, I believe they intend that most of the islets and rocks

are in the area (near) of the major islands.

Line 503: will be changed in "Consistent with Roe's suggestion"

Line 504: "sustains" will be changed in "maintains"

Line 538: will be change in "To Roe, these are clearly"

Line 545: ". . .with at great imagination." I believe that "imagery" is the right word.

Line 619: will be changed in " collected evidence"

Line 622: "WS respects.." Also, I do not really understand what this sentence about WS respecting the time, place, and action units of classic drama means; please clarify.

The sentence will be deleted. It regards the the classical unities of drama, but this is not relevant to my paper. What is relevant is that we can infer at what time the tempest takes place.

Line 624: Do you mean that the tempest that WS uses as a source must have occurred before the performance date of the play in 1611?

No I mean the tempest taking place in the play. From the conversation in verses I.ii 238-241, we can infer that the tempest occurs during the day and not at night.

Line 640: will be changed in "Fire is described..."

Line 714: will be changed in " and thought-executing "

Line 728: will be changed in "Eventually, WS was interested"

Line 729: will be changed in "rather than embarking upon"

Line 736: will be changed in "What has the Sea of Sicily to envy in the Bermuda triangle?"

Line 758: will be changed in "Channel has been an object of"

Line 767: I am not clear what this sentence starting "Eventually volcanic eruptions at sea..." means. Do you mean that it was probably known at the time of WS that sea eruptions occurred?

No. . . as I report from Mercalli (see 58-60) there are few records about volcanic eruptions at sea. So, I can only speculate that "volcanic eruptions at sea in this area may have occurred also at the age of WS". But I will try to make it more clear.

Line 771: Make sure to superscript the degree symbol, such as 36o I will do it

Line 850: Can you cite a source for this legend about Elizabeth I being thrown inside Aetna?

It is a legend born in Sicily. It is quoted also in a book of a Sicilian journalist S. Spoto. I will check it. . .

Line 875: will be changed in "Harpy interrupts the"

Line 882: will be changed in "collected evidence that"

Line 902: will be changed in "both fans of"

REFERENCES:

I will check references.

I stated the source for my quotes in 70

FIGURES:

Figure 2: Who generated this figure using Google Map – the author? If not provide the citation. Is using the face of WS the best symbol? Perhaps use coloured dots or crosses to avoid overlap?

I generated this map. . . yes, it is not a great map. I will try to make it better. . .

Figure 3 is charming, but is it necessary in this paper?

This article is part of a Special Issue on Earth sciences and Art, so I believe that we can add a figure that is not necessary in the paper. . .(I will discuss this with the Executive Editor)

Figure 4 is referred in the text in line 488. Should this be a reference to Figure 5?

I will correct this.

---

## Author Comment (AC2) · 25 Sep 2020

Thank you Anonymous Referee #2 for taking the time to review my article, and for some stimulating hints that surely will help improving my paper. Though, I should do some remarks that I will address in my answer.

**1 The topic broached by the author is a stimulating one, and I feel a genuine interest in the argument developed by the author. However, to my mind, the paper should be fully rewritten to be convincing and to appeal not only to specialists of geoscience, but also to Shakespeareans themselves. This is worth it. The first thing the author needs to consider is the length of her paper. It is much too long, all the more so as the first 12 pages or so seem more or less irrelevant and do not probe the issue of volcanism**

[Figure]

in the play.

MY ANSWER To #1 I totally agree that the paper is too long. I am not used to write long articles. In the specific case, the paper is conceived for a special issue devoted to Earth sciences and Art, so I had to address different audiences of expertise. The 12 pages that you consider more or less irrelevant have been written especially for literates (i.e. introduction; tempest storms and sea eruptions). I thought it would have been interesting for literates dealing with natural hazards to know a little bit more about the science behind a sea storm, and, in order to better understand my work, behind a sea eruption. Then, I wrote paragraphs especially addressed to the potential scientific readers. (par. 3.1; 3.2; 3.3). A paragraph is dedicated to the methodology I used to limit the area of interest, since, as we know, there are many interpretations of the The Tempest. The biography of WS is also subject to many controversies. Then, I believe that the text is the only reliable source worth to study, more than any other approach, and the interpretation of the verses, although difficult, can enlighten us on the possibility that WS was truly inspired by real places and by natural phenomena.

The paper will be surely shortened and will include some of your suggestions to appeal Shakespeareans too.

**2 Generalities should be removed, as well as confusing considerations on Shakespeare's authorship (Bacon, Florion), which have nothing to do with the scientific argument put forward here.**

MY ANSWER TO #2 I didn't discuss WS's authorship, even thought I consider this an important aspect to address to better understand his works. But I agree, I can remove what it is not immediately relevant to the present work.

**3 De facto, it contains too many factual errors and inaccuracies, especially regarding Shakespeare and the play itself. For instance, -The Tempest is NOT "considered the last play of WS mainly because it is a container of all the themes" previously dealt with by the playwright.**

MY ANSWER TO # 3 I put the argument in the wrong way. In writing so, I refer to some literature considering The Tempest as a reflection on the theatre itself (metatheatre) (see Frye 1986; Kernan 1979; Lombardo 1986;). In other words, Prospero is a playwright and a director, and, in this sense, an alter ego of WS. I didn't consider this an argument relevant to my paper, till the present discussion. The island, in this perspective, is a stage, where Prospero put into scene his own drama. There is a play within the play, and the borders between reality and illusion are extremely subtle. Even those between life and death (see the popular verses ""We are such stuff as dreams are made on, and our little life is rounded with a sleep.") This raises questions of fundamental importance for the text and its performance.

**4 The Tempest is NOT "the only play where WS respect the time, place and action units of classic drama". –**

MY ANSWER # 4 This is, actually, a specific argument not so relevant for my paper. What is relevant is that we can infer when the tempest takes place, from Ariel's words. The sentence will be deleted.

**5 Shakespeare's favorite setting is NOT the UK.**

MY ANSWER # 5 I didn't assert this in my paper. I provided a map of locations for his plays. UK and Italy are the places where he mostly located his works.

**6 Alonso's son is NOT named Ferdinando.**

MY ANSWER # 6 This is true. Ferdinando is the Italian for Ferdinand. I can easily correct this.

**7 The first storm may indeed be an illusion, which explains why the shipwrecked victims remain dry.**

This is not an argument. A storm is always an illusion when we are at theatre. The initial sea storm, is the "real storm", since it is the one that was put on stage. The audience don't know this is an illusion until the second scene of the play. Walking in

the shoes of WS, it must have been a work to convince the audience that the storm was an illusion, after the first scene, where we read "enter mariners wet" (see #3)

**8 But there are several storms on the island, and they are all treated differently (see Jones and Chiari for example).**

This is not correct. In the island there are not several storms. There are only two storms taking place. The one in the first scene (an illusion or not) (I.i vv 1-67) and a storm in the island (vv.II.ii vv38-42). All the rest, are narrations of past and contemporary events (the sea storm experienced by Prospero and Miranda twelve years before; The witnessing of the tempest by Miranda from the shore; Ariel reporting to Prospero about the tempest he provoked)

**9 Surprisingly, while the paper takes stock of recent trends in Shakespeare criticism, some fairly recent studies on ecocriticism (on Bacon in connection with The Tempest– see Popelard–and on storms, particularly) are never mentioned in this study, whereas obsolete works are repeatedly quoted. As a result, the bibliography should be updated with good references (Dan Falk, for example, cannot be said to be an absolute reference).**

MY ANSWER #9 I never consider "obsolete" past works. Although in scientific studies it would be more appropriate to address recent research, in a humanistic context, where demonstrating is such a hard task, past studies are still enlightening for many aspects. Nevertheless, I don't have anything against reading and considering further literature, although it would have been more helpful from your part indicating at least the year of publication in addition to surnames. But I agree, updating my references will help to address other important aspects not considered in the present draft and that will surely be addressed. Dan Falk is a science journalist, not an academic, but I simply quoted the excellent summary he does of the celestial events occurred in Shakespeare's time.

**10 Besides, the English should be checked and corrected by a native speaker: many sentences are awkward and the use of grammar is not always correct. For example:**

-[In recent years] a new trend towards the re-unification of the two main streams of culture, the humanistic and the scientific, is becoming more evident year by year. -Ovid's Metamorphosis (we should read Metamorphoses) -Archaeological and more recent remains found in the deep sea testify of a difficult navigation in dangerous water till present times. (we should read testify to) These mistakes are numerous and prevent a smooth reading of the text as a whole.

MY ANSWER # 10 The paper will surely benefit from a native speaker's linguistic revision.

**11 Methodologically, the author never relies on early modern translations, which is a problem. For we know that early modern translations were imperfect, and that there were important variants in the translated texts.**

MY ANSWER # 11 Relying on early modern translation cannot be my methodology, since I am not an expert in early modern translation. I have clearly indicated my methodology at the beginning of the paper. But I can surely refer to other studies in this sense. Nevertheless, it would have been more useful indicating how relying on early modern translations would help my paper. For instance, in the case of St. Elmo's fire I quoted sources contemporary to W.S. Hakluyt and, in particular, the Strachey's letter where the phenomenon of St. Elmo's fire is described. From this description it is immediately evident that Prospero couldn't have set up "a direful spectacle" inspired by this geophysical phenomenon.

**12 Moreover, which texts were translated in Shakespeare's time, and which were not? What access did he have to Strabo, for example?**

MY ANSWER # 12 Are these questions addressed in other papers? Have such studies already been performed? I will add, did Shakespeare know Greek and Latin to let us suppose that he didn't refer to second hand texts? This is not my task. But Again I can refer to some already existing literature in this sense. For instance, there is an interesting study by Werth, Andrew. "Shakespeare's 'Lesse Greek.'" The Oxfordian 5

(2002): 11-29 see here:

https://politicworm.files.wordpress.com/2009/04/werth-lesse-greek-tox022.pdf

This and more recent studies testify that the debate is on.

Concerning Strabo, I believe it is not absurd thinking that his Geography circulated at the time of Shakespeare. See for instance this article: Cormack, Lesley B. "Britannia Rules The Waves?: Images of Empire in Elizabethan England." Early Modern Literary Studies 4.2 / Special Issue 3 (September, 1998): 10.1-20 here: https://extra.shu.ac.uk/emls/04-2/cormbrit.htm#fn56

The importance of geographical studies for students is underlined in this article. In Fig.4 you can see the title page of a printed commonplace book where among the other personalities appears Strabo. Did Shakespeare had access to Strabo? I don't know, but I see similarities between some verses of the Tempest and the excerpt I quoted from his Geography. Maybe, it has come the time to do some research in this direction.

**13 Regarding mythology, the same problems crop up. Neptune was the god of earthquakes, but he was not particularly known as such in Shakespeare's times. It would therefore be crucial to study early modern representations rather than foregrounding our own perceptions of mythology.**

MY ANSWER #13 I believe it is a frail argument maintaining that since Neptune was not particularly known as the god of earthquakes in Shakespeare's time, then also Shakespeare wasn't aware of the double role of Neptune. In the Mediterranean mythology Neptune was also addressed as the Ennosigaeum (in Latin) the Earth shaker. The double role of Neptune is part of the ancient Mediterranean mythology. It all depends on whether we want to consider WS barely schooled in the classics or not (see # 12). In any case, I am here referring to mythology from a particular perspective as I have clearly indicated in my paragraph "The Tempest to the light of geo-mythology". In this

perspective, myths and legends have origin in the natural world, and can be seen as a source of natural knowledge based on the observation of physical evidence. So, I am not foregrounding my own perception of mythology. For instance, the interpretation of the trance-like state of the ancient Priestess, The Oracle at Delphi, as due to inhaling gas coming from a natural vent underneath the temple, is not only supported by scientific evidences, but also by ancient sources (Plutarch). (you can see the updating here: https://www.tandfonline.com/doi/full/10.1080/15563650701477803 In The Tempest, a state of mental confusion in some characters is reported after a natural event (being this an illusion or not). Then, one interpretation could be gas-inhaling as reported by Strabo.

\# 14 Finally, some premises seem particularly frail. Shakespeare probably knew sailors, yes. But how can that be proved? How do we know what he learnt from their testimonies?

MY ANSWER \# 14 These are not my premises. My premises are clearly indicated in par. 2 "It is our intention to collect all the indications that can help us to analyse all that in the play is connected to a real location in terms of an environmental and geophysical asset, using sources from geoscience studies, historical and others". In my article the fact that "WS could have had access to unknown sources as board diary of the vessel navigating those seas" is a hypothesis, since to this respect, as I have explained reporting Mercalli, there is a vulnus also in geophysical studies. Nevertheless, literary critics generally agree in indicating in Strachey's letter a source (that was not even published at the time he was writing The Tempest). Then, I don't see why WS shouldn't have considered the idea to find other "exotic" sources, especially if we think that the play indicates a very specific route (see Fig. 4)

\#15 More importantly, what did we know of the Vulcano islands (since Prospero's island would be partly inspired from this specific location), in the period? If Shakespeare knew about vulcanism, why does the author never quote any early modern text devoted to this particular phenomenon?

MY ANSWER #15 I didn't state that the island of The Tempest is Vulcano. I simply reports Roe's interpretation of some verses of the play. I have clearly stated since my abstract "We don't intend to identify the island". I respect the playwright's indication "An uninhabited island", although Volcano at the time of Shakespeare was an uninhabited island due to its volcanism, while as we read in Strachey, the Bermuda were wrongly considered uninhabitable. To my present study it is irrelevant to identify the island. In the same way, I didn't state that Shakespeare knew volcanism. Of course he didn't know about the causes of such phenomena. I believe he was familiar with volcanism as it appears directly from his verses. He may have read or heard about volcanic phenomena in the Mediterranean. Strabo is a source, these phenomena were observed since ancient times. Anyway an early modern text, J. Florio's dictionary World of Words, includes the definition of "Vulcani" as " always associated with fire". The dictionary includes also "Vulcanalia": "feastes dedicated to Vulcane"., ancient celebrations typical of the Mediterranean. In the dictionary we read also a description of St Elmo's fires: "Sant'E'rmo, taken for faire weather for Mariners, or prefaging of faire weather".

**16 In the play text, words such as sulphur and fire are present, but that is not enough to assert that Shakespeare describes a volcanic phenomenon.**

MY ANSWER #16 This could be true. I tried to analyse how WS uses words as "sulphurous", "fire" "roaring", and I have seen that in using "sulphurous" and "fire" he also refers to the sky. Nevertheless, volcanic phenomena were observed since ancient times, and no matter where or when, the descriptions of them are always similar since they have their origin in nature. In the second scene, the audience of the play, thanks to the words of Miranda, observes the tempest from the shore. She clearly says: "The sky, it seems, would pour down stinking pitch, But that the sea, mounting to th' welkin's cheek, Dashes the fire out" (Mir. I.ii. 3-4) How do you interpret these verses? She clearly says that "it seems" (may I think that it is not raining?), this is reinforced by "the sea ...dashes the fire out" where "fire" in this case is used with reference to the sky. She clearly says that "the sea mounts to the welkin's cheek". Then, it is a coincidence

that, as we read in Strabo, during a sea eruption the sea can mount "to an enormous height", in the same way it is described by Miranda's words? Today we know why this happens, you can relate to my par 3.5. from where you can also infer that there is a difference with sea storms. Not to mention "stinking pitch": you can clearly "smell" what is happening. And I believe it is not a case, if in The Tempest, we can literary "smell" the island. Finally, the timing, WS is so precise in this. When Miranda introduces the second scene she says: "If by your Art, my dearest father, you have Put the wild water in this roar, allay them."

From this verses we are able to understand that while Miranda speaks, the tempest is still in progress. Concerning the audience, the only difference from the first scene is the point of view. Now the audience is on the shore, observing what is happening into the sea, through Miranda's words. Secondly, the audience gets another description from Ariel's words. Comparing these two descriptions with what happens in the first scene, we note how different they are.

**17 (sulphur, as a matter of fact, related to hell in the early modern period, and it had also much to do with the pyrotechnics used for the stage–and that could have been a challenging argument)**

MY ANSWER #17 Sulphur related to hell not only in the early modern period, but also in Mediterranean mythology, and even today, because there is a reason: this imagery comes directly from volcanoes' landscapes. "That it had also much to do with the pyrotechnics used for the stage" can be a challenging argument for you and not for me. It could be interesting even intriguing, but to me is challenging to understand if words as "yellow" coupled with "sand" have only an aesthetic value or if they are descriptive of a real place. Or understanding what kind of pool is the one beyond Prospero's cell, and if WS when writing about "the foul lake" was inspired by a real landscape.

**18 He may very well describe, as has been argued elsewhere, a hurricane (and no need for that to rely on a wind imagery–suggestive images and evocative sentences**

were the very essence of early modern drama).

MY ANSWER #18 Hard to believe that the words of Ariel in vv. 193-206 I.ii describe a hurricane in an evocative manner. Even literates have searched for a geophysical explanation when they refer to St. Elmo's fire. The problem is that the very essence of St Elmo's fire is not fire. Likewise, the very essence of a sea storm or of a hurricane is not fire no matter how evocative a writer would like to be. This has been known since ancient times, because if it is true that there are few records about volcanic eruption at sea (as Mercalli reports) at the same time sea storms have been repeatedly described in similar ways in the literature worldwide.

**19 The paper actually heavily relies on Roe's arguments, but these have not always been regarded as convincing by Shakespeare critics: the analysis of the text remains too superficial to be enlightening.**

MY ANSWER # 19 My paper does not heavily relies on Roe's arguments. I presented this work at a conference in 2008 see: https://gsi.ir/en/articles/8455/sea-volcanism-in-sicily-and-mediterranean-myths-through-the-tempest-of-shakespeare three years before his book was published. I report his conclusions since the chapter in his book named "The Tempest, Island of wind and fire" as far as I know is the only one that leads some verses of The Tempest to volcanic landscapes. Unfortunately, he doesn't comment the verses I consider in par. 5.2. Saying that Roe's arguments have not always been regarded as convincing by Shakespeare critics is not enough. You should have made reference to studies disproving his conclusions.

Thank you for the interesting discussion. My paper will surely benefit from it.

---

## Author Comment (AC3) · 13 Nov 2020

Thank you again John Leigh, your suggestions have been precious. The article has been extensively reviewed to meet all the interesting ideas born from the discussion and is still in progress.

---

## Author Response (AR1)

I have been working on the paper to meet both the reviewers' suggestions. Some paragraphs have been eliminated to make space to arguments more in tune with the Shakespeareans as suggested by Rev #2. All the corrections suggested by Prof J.R.Leigh (REV #1) have been included apart where the text has been eliminated.
The minor revision of the Editor has been also included.

**The paper is now structured as follows:**

1) Introduction (shortened)

2) Methodology (long than before to meet Rev #2 requests)

2.1) Shakespeare and volcanoes (new to meet Rev #2 requests)

2.2) Tempests, storms and sea eruptions

3) Shakespeare's locations (shortened)

4) Introducing the tempest (includes a part of former "main streams of culture in English renaissance) The synopsis eliminated from the text, shortened and included in Fig. 3, together with the Dramatis Personae.

Somewhere beyond the sea: Shakespeare's sources for The Tempest, eliminated some part of it recuperated in the paragraph where Mercalli is treated.

4.1 The island of the Tempest: The Bermuda Hypothesis (shortened)

4.2 Echoes of the Bermuda in The Tempest

4.3 Placing the island into a Mediterranean context (part of it cut and shortened and put in par. 5.2)

5. Natural hazards in the Tempest: a fire-based play

5.1 A tempest or a sea eruption?

5.2 Volcanism in the Sicilian Sea and The Tempest

5.3 The Tempest in the light of geo-mythology

Conclusion

References and Figs have been re-organized according Rev #1 suggestions.
Tab. 1 has been added

**I include the paper with the new parts marked in yellow.**

[revised manuscript text omitted]